# Measurement report: Summertime and wintertime VOCs in Houston: Source apportionment and spatial distribution of source origins

Bavand Sadeghi[1], Arman Pouyaei[1], Yunsoo Choi[1*], Bernhard Rappenglueck[1]

[1]Department of Earth and Atmospheric Science, University of Houston, 77054, USA

*Correspondence to*: Yunsoo Choi (ychoi6@uh.edu)

**Abstract**: The seasonal variations of volatile organic compounds (VOCs) was studied in the Houston metropolitan area in the summertime and wintertime of 2018. The analysis of hourly measurements obtained from the automated gas chromatograph (auto-GC) network showed the total VOC average concentrations of 28.68 ppbC in the summertime and 33.92 ppbC in the wintertime. The largest contributions came from alkane compounds, which accounted for 61% and 82% of VOCs in the summer and winter, respectively. We performed principal component analysis (PCA) and Positive Matrix Factorization (PMF) and identified seven factors in the summertime and six factors in the wintertime, among which alkane species formed three factors according to their rate of reactions in both seasons: (1) the emissions of long-lived tracers from oil and natural gas (ONG long-lived species), (2) fuel evaporation, and (3) the emissions of short-lived tracers from oil and natural gas (ONG short-lived species). Two other similar factors were (4) emissions of aromatic compounds and (5) alkene tracers of ethylene and propylene. Summertime factor 6 was associated with acetylene, and one extra summertime factor 7 was influenced by the biogenic emissions. The factor 6 of wintertime was affected by vehicle exhaust. Higher nighttime and lower daytime values of the ethylene/acetylene ratios during the summertime indicated the stronger impacts of ethylene photochemical degradation. Also, the exploration of the photochemical processes of the VOCs showed that the ethylene and propylene had the highest contributions to the summertime and wintertime ozone formation as well as the emissions of the isoprene, which showed a high impact on summertime ozone. Our results acknowledged that ethylene and propylene continue to be significant emissions of VOCs, and their emissions control would help the mitigation of the ozone of Ship Channel. Based on trajectory analysis, we identified main VOC emission sources in Houston Ship Channel (HSC) local industrial areas and regions south of the HSC. Ambient VOC concentrations measured at the HSC were influenced by the emissions from the petrochemical sectors and industrial complexes, especially from the Baytown refinery and Bayport industrial district next to the HSC and Galveston Bay refineries at the south of the study area.

**Keywords:** Volatile organic compounds, Seasonal variation, Source apportionment, Trajectory analysis, ozone photochemistry

## 1 Introduction

Volatile organic compounds (VOCs) encompass a variety of carbon-containing compounds with various lifetimes from hours to months (Atkinson, 2000). The direct and indirect impacts of VOC compounds on humans and the environment cause a number of adverse health effects. Diseases caused by the direct exposure to VOCs include a wide range of serious symptoms from eye, nose, and throat irritation to damage to the liver, kidney, and nervous system, neurological toxicity, and lung cancer (Burton and B., 1997; Kim et al., 2019; Maroni et al., 1995; Mølhave, 1991; World Health Organization., 2000). Their sensitivity to photochemical reactions with radicals (e.g., OH, $O_3$, Cl) and nitrogen oxides ($NO_x$) in the presence of sunlight leads to the formation of ground-level ozone, which exacerbates chronic conditions in humans and harms agricultural products and natural ecosystems (Carter, 1994; Chameides et al., 1992; Liu, 1987; Logan et al., 1981; Mousavinezhad et al., 2021). The gas-to-particle partitioning of VOC compounds can lead to the formation of secondary particulate matter (Heald et al., 2020; Izumi and Fukuyama, 1990; Odum et al., 1997) that affects atmospheric radiative forcing and the climate (Ng et al., 2007; Tsigaridis and Kanakidou, 2003) and is associated with critical health impairments (Nault et al., 2020). Any strategy for emission mitigations of VOCs, however, necessitates a more comprehensive understanding of the source-receptor relationships, including advanced atmospheric transport algorithms.

Atmospheric VOCs originate from emissions of either anthropogenic or natural biogenic sources. On a global scale, natural emissions contribute more than 80% of total emissions. Biogenic VOCs emissions originate from biomass burning and complex gas-phase exchange processes in land-based plants dependent on temperature and light (Forkel et al., 2006; Fuentes et al., 2000; Karl et al., 2003; Kesselmeier and Staudt, 1999). In urban areas, however, numerous anthropogenic sources emit VOCs into the atmosphere and likely exceed biogenic emissions in these areas (Rappenglueck et al., 2005; Winkler et al., 2002). The main sources of anthropogenic emissions are industrial processes, including those of crude oil and liquefied petroleum gas (LPG), hereinafter defined as oil and natural gas (ONG), gasoline transport and storage, vehicle combustion, and the manufacturing production of commercial goods (Piccot et al., 1992). A number of studies have identified these VOC sources, their emission strengths and contributions in photochemical processes in several urban areas, including Houston (Bi et al., 2021; Buzcu and Fraser, 2006; Czader et al., 2008; Czader and Rappenglueck, 2015; Diao et al., 2016; Jobson et al., 2004; Leuchner and Rappenglueck, 2010; Pan et al., 2015). These studies reviewed the effectiveness of control strategies and policy regulations and links between VOC concentration levels and surface ozone production as explored by Kleinman et al. (2002).

Houston has been the site of a number of studies characterizing the emissions of organic compounds and their impact on ozone formation in urban areas. The Houston metropolitan area has some of the largest anthropogenic emission sources of atmospheric pollutants in the United States (Song et al., 2021). The Houston Ship Channel area, for example, is affected by high rates of numerous pollutants from petrochemical and industrial facilities (Port Houston, 2019). Buzcu and Fraser showed that refinery and petrochemical activities are the dominant sources of VOCs at three sites in the Ship Channel area of Houston (Buzcu and Fraser, 2006). Building upon VOC source apportionment for six sites in the Houston's Ship Channel area from August to September of 2006, (Leuchner and Rappenglueck, 2010) expanded that analysis further by looking into diurnal variations and including wind directional dependencies for a

receptor site outside the industrial region of the Ship Channel which was also exposed to urban emissions, and attributed the contributions of anthropogenic and biogenic emissions to ambient VOCs on the site.

This study aims to explore the characteristics of VOCs (such as concentration levels, diurnal variations, and emission features) and examine their seasonal variations for the emission factors of the Houston Ship Channel based on the measurements of the summertime and wintertime 2018. In the present paper, we document the study domain and measurement datasets as well as the methods of our study (Sect. 3.1), which includes an integrated source-receptor relationship study for the Houston Ship Channel region using Principal Component Analysis (PCA) and Positive Matrix Factorization (PMF) models to characterize the emissions of VOCs and their temporal variation in this industrial sector of the Houston metropolitan area in summer- and wintertime (Sect 3.2). We also discuss the removal process of the VOC and their contributions to ozone formation (Sect. 3.3). Leuchner and Rappenglueck (2010) provided a categorical view about the wind direction frequency for emission sources, but this paper goes beyond by studying the geographical origins of the air masses coming from the larger Houston metropolitan area (Sect 3.4).

## 2 Materials and Methods

### 2.1 Site description and measurements

For this study, we used in-situ measurements of VOCs for the summertime and wintertime of 2018 in Houston industrial regions from Photochemical Assessment Monitoring Stations (PAMS) operated by the Texas Commission on Environmental Quality (TCEQ) auto-GC system. These datasets were reported by Texas Air Monitoring Information System (TAMIS), and the analytical method and procedure for method detection limits (MDLs) are available at EPA technical air pollution resources. The summer season is defined as June, July, and August (JJA) and the winter season as December, January, and February (DJF). Among the auto-GC sites, data from Channelview, HRM #3 Haden Road, Clinton, and Lynchburg Ferry was compiled for this study (Fig. 1).

The Channelview site is located on the northeastern curve of the Houston Ship Channel at the east of Downtown Houston. Haden Road site is on the top of Houston Buffalo Bayou at the east of Channelview. Clinton site is located near Port of Houston surrounded by both residential and industrial areas, and the Wallisville Road site is near Baytown city on the road to an exurb northeast of Houston. The Lynchburg Ferry site is located in a park and recreation area on a peninsular stretching into the HSC, exposed to a complex mixture of various emission sources across the Houston Ship Channel. For experimental details, see Leuchner and Rappenglueck (2010) and the latest update by the US Environmental Protection Agency (EPA, 2019).

Atmospheric Chemistry and Physics
EGU
Discussions

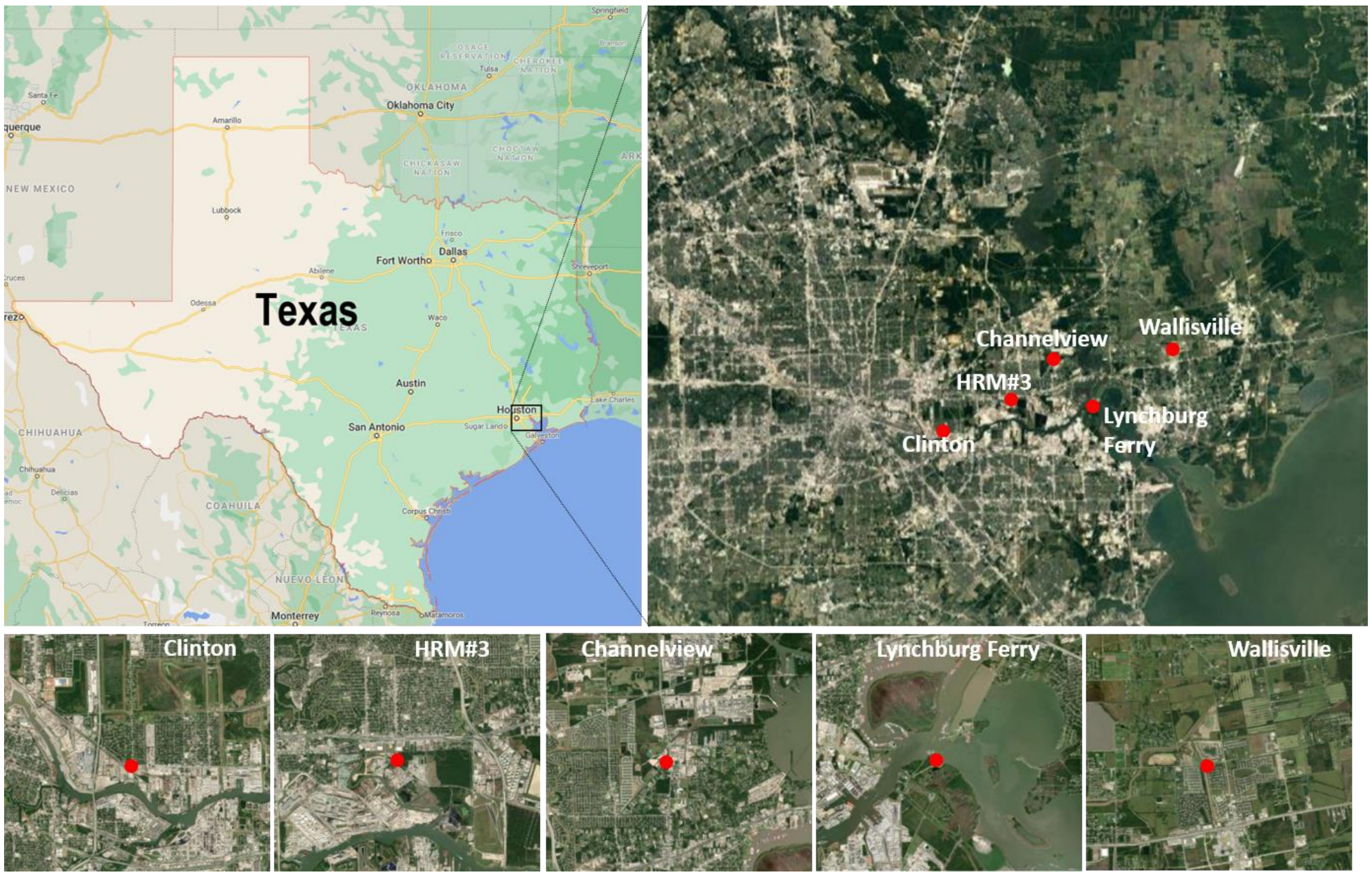

**Figure 1.** Geographical locations of the five VOC auto-GC sites (Clinton, HRM#3, Channelview, Lynchburg Ferry, and Wallisville) in Houston (from © Google Maps).

**Table 1.** Chemical Group, OH Rate Coefficients, Average Mixing Ratios, Standard Deviations, and Medians for selected VOCs at the Lynchburg Ferry site during summertime and wintertime 2018.

| VOC | Chemical Group | $10^{12}\times$ $K_{OH}$[a] | MIR[b] | Units: ppbC | | | | | |
|---|---|---|---|---|---|---|---|---|---|
| | | | | Mean | | Standard Deviation | | Median | |
| | | | | summer | winter | summer | winter | summer | winter |
| Ethane | alkane | 0.248 | 0.28 | 4.66 | 10.76 | 4.38 | 7.41 | 3.41 | 8.93 |
| Ethylene | alkene | 8.52 | 9.00 | 4.19 | 1.71 | 10.09 | 3.65 | 1.09 | 0.71 |
| Propane | alkane | 1.09 | 0.49 | 3.11 | 6.74 | 4.51 | 6.77 | 1.69 | 5.01 |
| Propylene | alkene | 26.3 | 11.66 | 4.00 | 1.76 | 15.87 | 7.29 | 0.50 | 0.35 |
| Acetylene | alkyne | 0.815 | 0.95 | 0.19 | 0.32 | 0.47 | 0.33 | 0.15 | 0.25 |
| *n*-Butane | alkane | 2.36 | 1.15 | 1.93 | 3.42 | 3.12 | 5.80 | 0.96 | 2.13 |
| *iso*-Butane | alkane | 2.34 | 1.23 | 1.79 | 2.02 | 4.36 | 5.01 | 0.73 | 1.06 |
| *trans*-2-Butene | alkene | 64.0 | 15.16 | 0.06 | 0.06 | 0.11 | 0.15 | 0.08 | 0.03 |
| *cis*-2-Butene | alkene | 56.4 | 14.24 | 0.08 | 0.05 | 0.13 | 0.12 | 0.09 | 0.03 |
| 1.3-Butadiene | alkene | 66.6 | 12.61 | 0.10 | 0.12 | 0.34 | 0.47 | 0.09 | 0.03 |
| *n*-Pentane | alkane | 3.8 | 1.31 | 1.52 | 1.26 | 2.72 | 2.20 | 0.63 | 0.66 |
| *iso*-Pentane | alkane | 3.9 | 1.45 | 2.23 | 1.84 | 3.76 | 4.73 | 0.93 | 0.86 |
| 1-Pentene | alkene | 31.4 | 7.21 | 0.05 | 0.04 | 0.13 | 0.11 | 0.03 | 0.02 |
| *trans*-2-Pentene | alkene | 67 | 10.56 | 0.04 | 0.03 | 0.09 | 0.11 | 0.03 | 0.02 |
| *cis*-2-Pentene | alkene | 65 | 10.38 | 0.02 | 0.01 | 0.05 | 0.07 | 0.03 | 0.02 |
| n-Hexane | alkane | 5.2 | 1.24 | 1.11 | 0.57 | 2.08 | 1.36 | 0.50 | 0.26 |
| n-Octane | alkane | 8.11 | 0.90 | 0.07 | 0.10 | 0.11 | 0.26 | 0.04 | 0.04 |
| n-Nonane | alkane | 9.7 | 0.78 | 0.03 | 0.05 | 0.04 | 0.11 | 0.02 | 0.02 |
| n-Decane | alkane | 11.0 | 0.68 | 0.04 | 0.06 | 0.13 | 0.21 | 0.02 | 0.02 |
| Cyclopentane | alkane | 4.97 | 2.39 | 0.15 | 0.13 | 0.18 | 0.36 | 0.08 | 0.07 |
| Isoprene | alkene | 101 | 10.61 | 0.47 | 0.06 | 3.56 | 0.25 | 0.31 | 0.03 |
| 2.2-Dimethylbutane | alkane | 2.23 | 1.17 | 0.03 | 0.03 | 0.07 | 0.06 | 0.03 | 0.02 |
| 2.4-Dimethylpentane | alkane | 4.77 | 1.55 | 0.02 | 0.02 | 0.10 | 0.24 | 0.27 | 0.02 |
| 3-Methylhexane | alkane | *N/A* | 1.61 | 0.08 | 0.08 | 0.13 | 0.22 | 0.09 | 0.06 |
| 2.2.4-Trimethylpentane | alkane | 3.34 | 1.26 | 0.18 | 0.29 | 0.36 | 2.18 | 0.11 | 0.08 |

| | | | | | | | | | |
|---|---|---|---|---|---|---|---|---|---|
| 2.3.4-Trimethylpentane | alkane | 6.6 | 1.03 | 0.03 | 0.07 | 0.05 | 0.90 | 0.03 | 0.02 |
| 3-Methylheptane | alkane | *N/A* | 1.24 | 0.02 | 0.04 | 0.04 | 0.10 | 0.02 | 0.03 |
| Methylcyclohexane | alkane | 9.64 | 1.70 | 0.13 | 0.19 | 0.19 | 0.50 | 0.08 | 0.08 |
| Methylcyclopentane | alkane | 7.04 | 2.19 | 0.39 | 0.26 | 0.65 | 0.83 | 0.20 | 0.11 |
| 2.3-Dimethylpentane | alkane | *N/A* | 1.34 | 0.03 | 0.03 | 0.08 | 0.19 | 0.17 | 0.19 |
| 2-Methylheptane | alkane | *N/A* | 1.07 | 0.03 | 0.05 | 0.05 | 0.15 | 0.03 | 0.02 |
| *m.p*-Xylene | aromatic | 23.1 | 7.79 | 0.20 | 0.22 | 0.34 | 0.46 | 0.09 | 0.08 |
| Benzene | aromatic | 1.22 | 0.72 | 1.11 | 0.63 | 2.57 | 1.61 | 0.29 | 0.18 |
| Toluene | aromatic | 5.63 | 4.00 | 0.39 | 0.45 | 0.58 | 1.65 | 0.20 | 0.19 |
| Ethylbenzene | aromatic | 7.0 | 3.04 | 0.05 | 0.06 | 0.08 | 0.17 | 0. 03 | 0.03 |
| *o*-Xylene | aromatic | 13.6 | 7.64 | 0.05 | 0.06 | 0.09 | 0.14 | 0.03 | 0.03 |
| 1.3.5-Trimethylbenzene | aromatic | 56.7 | 11.76 | 0.01 | 0.02 | 0.03 | 0.06 | 0.02 | 0.02 |
| 1.2.4-Trimethylbenzene | aromatic | 32.5 | 8.87 | 0.03 | 0.06 | 0.07 | 0.14 | 0.02 | 0.03 |
| n-Propylbenzene | aromatic | 6.0 | 2.03 | 0.01 | 0.02 | 0.02 | 0.03 | BMDL | 0.02 |
| Isopropylbenzene | aromatic | 6.5 | 2.52 | 0.02 | 0.01 | 0.12 | 0.04 | 0.02 | BMDL |
| Styrene | aromatic | 58 | 1.73 | 0.03 | 0.22 | 0.20 | 2.03 | 0.03 | 0.04 |

[a] Rate constant of NMHCs react with OH at 298 K ($cm^3$ $molecule^{-1}$ $s^{-1}$) (Atkinson and Arey, 2003)
[b] MIR denotes maximum incremental reactivity (Carter, 1994)
[c] BMDL denotes Below Method Detection Limits
[*] N/A: No literature $K_{OH}$ value available.

Out of 48 compounds measured by the auto-GC system, seven species 2-methyl-2-butene, n-heptane, cyclohexane, 2-methylhexane, 1-butene, n-undecane, and 1.2.3-Trimethylbenzene had to be excluded from the datasets due to their smaller signal/noise ratio or they had > 25% of missing values.

A list of alkane, alkene, alkyne, and aromatic tracers for this study is shown in Table 1, here exemplary for the Lynchburg Ferry site. Based on the average values of the summertime and wintertime concentrations in this Table, these measurements were dominated by alkanes, followed by alkenes and aromatics.

The concentrations of each VOC class changed from station to station (Fig. 2). The total concentration of summertime VOC level was highest at Lynchburg Ferry and HRM #3 Haden Road. But the measurements of alkane, alkene, and aromatic compounds show higher concentrations during the wintertime. At almost all sites, alkanes were significantly higher in wintertime compared to summertime. The plots of alkenes and aromatics, however, showed less seasonal differences in concentrations than alkanes, albeit they have a shorter atmospheric lifetime. The alkane concentrations differences between the summertime and wintertime could be explained by the longer lifetimes of ethane and propane, the two most abundant alkanes. Assuming limited seasonal changes in emissions, the more rapid reaction of the alkenes and aromatics with radicals would cause their seasonal differences less variant. Because of the higher average concentrations of alkenes and aromatic compounds at the Lynchburg Ferry station and the higher number of PCA-resolved factors during both summer and winter 2018 (Table S2), along with the strategic location of the site amidst a crucial petrochemical region in the United States, we analyzed and characterized the measured VOCs at that site.

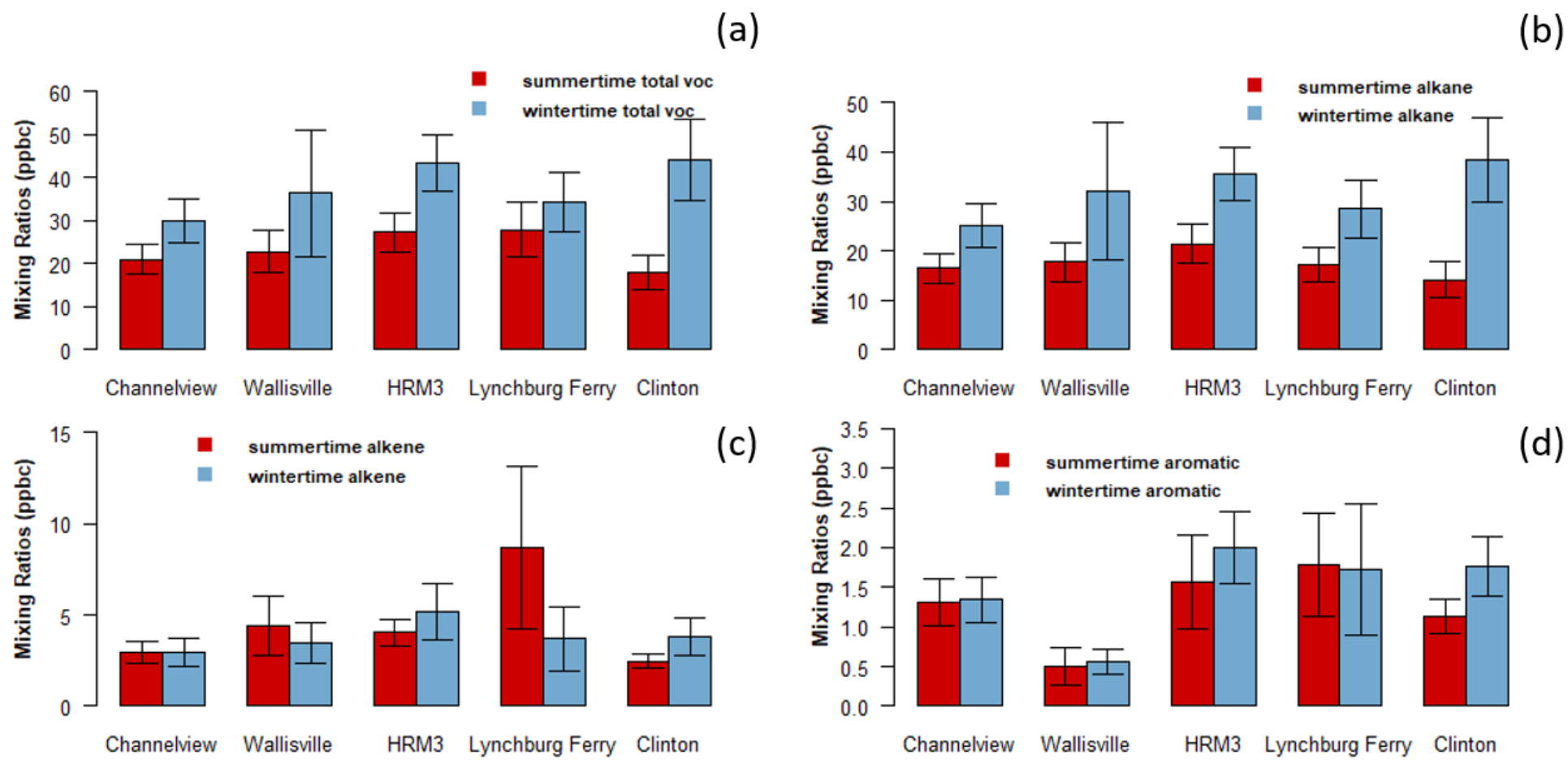

**Figure 2.** Box plots of hourly concentrations of VOC classes at the different auto-GC stations. (a) total VOCs, (b) alkanes, (c) alkenes, and (d) aromatics.

## 2.2 Factor analysis models

### 2.2.1 Principal component analysis (PCA)

A multivariate statistical technique, PCA transforms an original group of inter-correlated variables into a new group of independent and uncorrelated variables (Henry and Hidy, 1979; Sadeghi et al., 2020). To perform the PMF, we used the PCA method to reduce the original number of samples into the required number of factors. For the factor analysis, we used the Kaiser-Meyer-Olkin (Kaiser, 1970) and Bartlett's tests to evaluate and confirm the adequacy and sphericity of the measured dataset.

### 2.2.2 Positive matrix factorization (PMF)

Receptor modeling is a source apportionment technique that elicits information about the sources of air pollutants (Hopke, 2003). Receptor models are based on the application of a mass balance analysis on measured mass concentrations (Hopke, 2016). Based on mass conservation, the PMF receptor model is an advanced multivariate factor analysis tool commonly used to identify and quantify the contributions of primary sources of ambient measurements. The mathematical theory and principles of the model were extensively elaborated by Paatero and Tapper (1994).

This statistical model consists of decomposing the matrix of measured ambient data into two factor profile matrices and contributions. Eq (1) briefly defines this principle:

$$x_{ij} = \sum_{k=1}^{p} g_{ik} f_{kj} + e_{ij}, g_{ij} \geq 0, f_{kj} \geq 0 \quad (1)$$

where $x_{ij}$ represents ambient data concentrations, in which $i$ the number of samples and $j$ chemical species were measured, $g_{ik}$ is the factor contribution of the $k^{th}$ factor to the $i^{th}$ sample, $f_{kj}$ is the factor profile of the $j^{th}$ species in

the $k^{th}$ factor, and $e_{ij}$ represents the residuals that the model cannot fit. The goal of PMF is to find the non-negative matrices of the factor contribution and profile that lead to the minimum value of the objective function Q, defined as Eq (2), where $u_{ij}$ is an estimate of the uncertainty in chemical species $i$ of the sample $j$ (Paatero, 1999):

$$Q = \sum_{i=1}^{n}\sum_{j=1}^{m}\left[\frac{e_{ij}}{u_{ij}}\right]^2 \quad (2)$$

A weighted least square method of PMF minimizes the differences between the measurement values and the model-calculated data. This algorithm constrains all of the matrix components of the factor contribution and profiles to ≥ 0. We have used the EPA PMF 5.0 to identify the sources of VOCs measurements (Norris et al., 2014).

For the uncertainty of the species, we compared the concentration values to the method detection limit (MDL). If the concentrations were lower than or equal to the MDL, uncertainty calculation was based on Eq. (3):

$$Unc = \frac{5}{6} \times MDL \quad (3)$$

If the species concentration were higher than the MDL, based on the concentration, MDL, and the error fraction (Norris et al., 2014), the uncertainty would be estimated by Eq. (4):

$$Unc = \sqrt{(Error\ Fraction\ \times concentration)^2 + (0.5 \times MDL)^2} \quad (4)$$

### 2.3 Ozone formation potential

Tropospheric ozone production depends on the availability of $NO_x$ and VOC. The contribution of individual VOCs to ozone formation largely depends on their concentration and their reactivity with the hydroxyl radical, which both cover a wide range (Table 1). The ozone formation potential (OFP) is an ozone sensitivity indicator that describes the relative effects of VOCs on ozone formation in the troposphere and identifies key sources and species for ozone formation. OFP scales have been widely used to evaluate the roles of different organic compounds in ozone formation (Kumar et al., 2020; Li et al., 2019; Marvin et al., 2021; Xie et al., 2008). In this study, we used two methods, i.e., the propylene-equivalent weighted concentration and maximum incremental reactivity MIR-weighted concentration, to evaluate the $O_3$ formation potential by VOCs kinetic reactivity and mechanism reactivity, respectively (Duan et al., 2008). The propylene-equivalent method estimates the OFP of each VOC based on its kinetic reactivity with hydroxyl radical normalized against the corresponding reactivity of propylene (Chameides et al., 1992); reaction mechanisms between VOCs, peroxide radicals, and NO are ignored (Zou et al., 2015). The calculation of the propylene-equivalent concentration for each VOC is shown in Eq. (5):

$$C_{j,propy-Equiv} = conc_j \times \frac{K_{OH}(j)}{K_{OH}(Propylene)} \quad (5)$$

where $C_{j,propy-Equiv}$ denotes the propylene-equivalent concentration (ppbC) of species $j$, $Conc(j)$ represents the carbon atom concentration (ppbC) of species $i$, $K_{OH}(j)$ and $K_{OH}(Propylene)$ denotes the chemical reaction rate (cm$^3$ molecule$^{-1}$ s$^{-1}$) of species $j$ and propylene with OH radical at 298 K, respectively (Table 1). The MIR-weighted

concentration, proposed by Carter (1994), considers the impacts of different reaction mechanisms and VOCs/$NO_x$ ratios on ozone formation. Therefore, it represents both the reactivity of individual VOCs to oxidation by OH radicals and the capacity of these VOCs to ozone formation as follows (Atkinson, 2000):

$$C_{j,MIR} = MIR_j \times C_{j,ppbv} \times \frac{m_j}{M} \quad (6)$$

where $C_{j,ppbv}$ is the ozone production potential for species $j$, $M$ represents the molecular mass of ozone, and $m_j$ represents the relative molecular mass of species $j$ in the VOCs. The $MIR_j$ is the maximum incremental reactivity coefficient of $j^{th}$ VOC. These values were estimated by selecting the MIR value for each of the VOCs from modeled scenarios conducted for Los Angles in 1980 (Carter, 1994; Dodge, 1984). Here, we use the most recent MIR values for VOCs provided by Carter (2010), defined as grams of ozone formed per gram of VOC emitted. The OH reaction rate constants and MIR coefficients for 41 VOCs are shown in Table 1.

**2.4 Determination of the concentration weighted trajectory (CWT)**

To localize potential regional sources of measured hydrocarbons at the auto-GC sites in the Houston Ship Channel, we used the CWT algorithm, which is a function of VOC concentrations reported every 2 hours, and the residence time of the trajectory crossing at each grid cell. We prepared one trajectory every measurement hour for the summertime and wintertime of 2018. Prior to using the CWT, we acquired back trajectories for each two hour measurement of VOCs by the National Oceanic and Atmospheric Administration (NOAA) Hybrid Single-Particle Lagrangian Trajectory (HYSPLIT) model (Stein et al., 2015). The 12 back trajectories with two hour intervals, starting from 00:00 to 22:00 (Central Standard Time), were derived from the Global Data Assimilation System wind field data archive from the National Weather Service National Center for Environmental Prediction. To investigate the impacts of turbulence created by hot emission plumes at the site, we considered varying boundary layer heights. As shown in Fig. S4, CWT results did not change for different altitudes. Thus, the arrival height was set as 100 meters above ground levels, the height at which the impact of turbulence created by hot emission plumes is minimized. In the following equation (5), each grid cell received a weighted concentration by averaging the sample concentrations of each of the PMF-resolved factors with associated trajectories that passed that grid cell as follows:

$$CWT_{ij} = \frac{\sum_{l=1}^{L} C_l \tau_{ijl}}{\sum_{l=1}^{L} \tau_{ijl}} \quad (7)$$

where $C_{ij}$ represents the weighted average concentration in a grid cell $(i, j)$, $C_l$ is the measured concentration at the sampling site during day $l$, and $\tau_{ij}$ is the residence time of the 24 backward trajectories corresponding to the day $l$in grid cell $(i, j)$ (Liu et al., 2021).

## 3. Results and Discussion

### 3.1 Data overview and general characteristics of VOCs
The statistics of 41 organic compounds that we analyzed at the Lynchburg Ferry station are shown in Table 1. The total average VOC concentration at this station in the summertime was 28.68 ppbC, consisting of 17.57 ppbC of alkanes; 8.54 ppbC of alkenes, 1.90 ppbC of aromatics, less than 0.2 ppbC of alkyne, and 0.47 ppbC of isoprene. The average concentrations of wintertime VOCs showed higher values: 33.88 ppbC comprised of 27.98 ppbC of alkanes, 3.77 ppbC of alkenes, 1.74 ppbC of aromatics, 0.32 ppbC of alkyne, and 0.06 ppbC of isoprene. Ethane and propane showed the greatest differences in VOC concentrations between the seasons. These two alkanes together formed 27% and 52% of total VOCs in the summertime and wintertime, respectively.

With regard to the differences between the concentrations of measured compounds in summer and winter, there are several factors such as emission sources, convective mixing, and photochemical reactions which caused the seasonal variations. With regard to the emission sources, an annual peak reflecting fluctuations in energy demand occurs during the summer months with the increased use of electricity for air conditioning. Higher daytime convective boundary layer due to turbulence likely led to reduced summertime concentrations due to vigorous turbulent mixing and dilution compared to wintertime conditions. With regards to the photochemical reactions, stronger solar radiation and higher temperatures accelerate the photochemical reaction processes of VOCs, mostly by hydroxyl radicals (Atkinson et al., 1997). These processes result in stronger removal of VOCs during the day in the summertime. Altogether, these factors contribute to lower VOC concentrations in the summer than in the winter.

### 3. 2. Factor profiles by the PMF analysis

The PCA model for the Lynchburg Ferry site indicated seven and six factors, in the summer and winter, respectively. The results of the PMF modeling reveal the sources of measured VOCs in the Houston Ship Channel in 2018 (Fig. 3). The seven summertime factors were (1) ONG long-lived species, (2) fuel evaporation, (3) ONG short-lived species, (4) aromatic, (5) ethylene and propylene, (6) acetylene, and (7) biogenic emissions (Fig. 3a). A PMF six-factors were resolved for the winter VOCs data-set (Fig. 3b). These factors were (1) ONG long-lived species, (2) fuel evaporation, (3) ONG short-lived species, (4) aromatic, (5) ethylene and propylene, and (6) vehicular exhaust.

The PMF algorithm treats the measured organic compounds as inert species, which adds some limitations to the interpretation of the results. These limitations are due to the simplification of processes such as mass production and atmospheric removal, e.g., loss of organic compounds through a chemical reaction, formation of secondary organic aerosols, and dry deposition. The method, however, provides supportive and reliable information concerning the local emissions of these compounds by classifying the VOCs into a group of factors or source profiles and giving each factor different weights of VOCs that represent the individual emission source. These source profiles or factors are assigned according to the current knowledge of the VOCs fingerprints of emission sources.

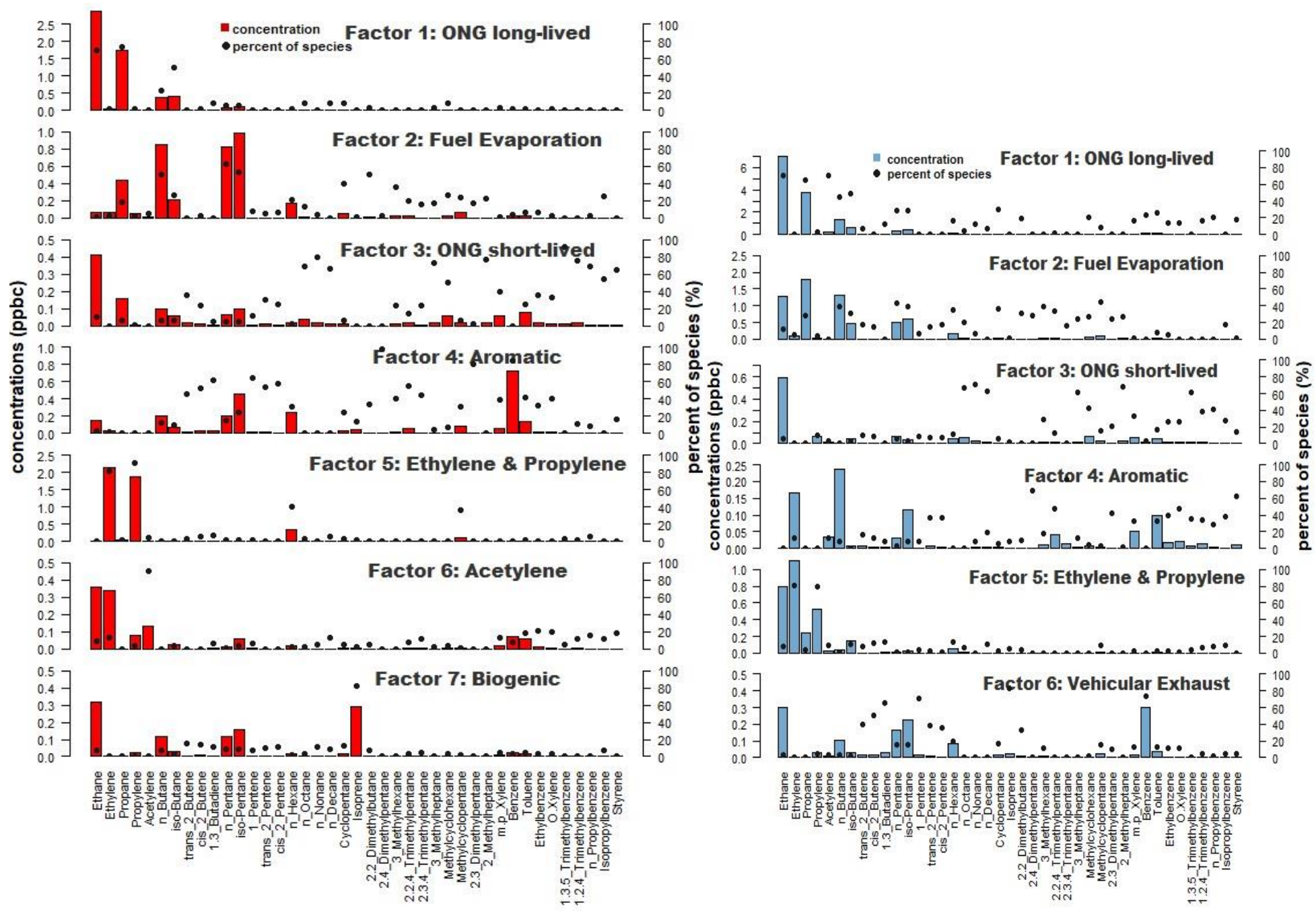

**Figure 3.** PMF factor profiles for (a) the summertime (left) and (b) the wintertime (right) at the Ship Channel Lynchburg Ferry site. Summertime factors (1) to (7) are ONG long-lived, fuel evaporation, ONG short-lived species, aromatic, ethylene and propylene, acetylene, and biogenic emissions. Wintertime factors (1) to (6) are ONG long-lived species, fuel evaporation, ONG short-lived species, vehicular exhaust, aromatic, and ethylene and propylene emissions. The concentrations (ppbC) and the percentage of each species (%) apportioned to the factors are displayed as colored bars and black points, respectively.

As shown in Fig. 3, three resolved factors of VOCs turned out to be shaped by the high presences of alkanes. Factors 1 and 3 were consistent with the emissions of ONG-related hydrocarbons. The first factor, ONG long-lived species, was more dominated by less-reactive species, whereas the third factor, with more highly-reactive species, was identified as ONG short-lived species. The factor ONG long-lived species was comprised of the high mixing ratios of simple alkane compounds. This factor contains ethane (69.0% and 66.3%) and propane (73.9% and 58.2%) for the summertime and the wintertime, respectively. Because of the high abundances of ethane and propane, this factor represents anthropogenic fossil fuel emissions, which are the primary contributors of these two hydrocarbons, according to most global inventories (Dalsøren et al., 2018; Koppmann, 2008). Previous studies have indicated the contributions of fossil fuel emissions to elevated ambient levels of ethane and propane in Houston (Buzcu and Fraser, 2006; Jobson et al., 2004). Our findings indicate that the ONG long-lived species played a significant role in both seasons, and it is unlikely that other anthropogenic sources contributed to ethane and propane emissions at the Lynchburg Ferry site.

In the summertime, the VOC composition of factor 2, which includes *n*-butane (51.1%), *iso*butane (26.7%), *n*-pentane (62.8%), and *iso*-pentane (52.0%) of their total percentages, hints at fuel evaporation in line with earlier findings (Leuchner and Rappenglueck, 2010). By comparison, in the wintertime, *n*-butane, *iso*-butane, *n*-pentane, and *iso*-

pentane accounted for 46.5%, 39.4%, 49.7%, and 44.2% of fuel evaporation, respectively. Factor 3, ONG-short-lived species, was dominated by alkanes with higher C8-C10 isomers. This factor includes 2-methylheptane (77.2% and 68.1%), 3-methylheptane (72.7% and 61.7%), n-octane (69.2% and 66.6%), n-nonane (80.1% and 72.4%), and n-decane (65.8% and 64.6%) for the summertime and wintertime, respectively. The emissions of these alkanes have been linked to operations associated with crude oil and condensate tanks (Berger and Anderson, 1981; Warneke et al., 2014). While the alkane-based factors, —ONG long-lived, fuel evaporation, and ONG short-lived factors— are associated with the emissions related to petrochemical industries and natural gas, these three PMF-resolved factors could be classified and distinguished in terms of their photochemical lifetimes. In the summertime, the contributions of these factors were 27.2%, 19.2%, and 6.7% of total measurements for ONG long-lived, fuel evaporation, and ONG short-lived species, respectively. In the wintertime, the contributions of these factors were higher (50.9%, 24.2%, and 4.8%, respectively).

The OH reaction rate constants of C2 (ethane) and C3 (propane), the signatures of the ONG long-lived factor, are around the order of (0.25-1.0) × $10^{-12}$ $cm^3$ $molecule^{-1}$ $s^{-1}$. The signatures of the fuel evaporation factor, isomers of C4-C5 isomers, have OH reaction rate constants of about (2.5-4.0) × $10^{-12}$ $cm^3$ $molecule^{-1}$ $s^{-1}$. For ONG short-lived species, C8-C10 isomers have OH reaction rate constants of about (8-11) × $10^{-12}$ $cm^3$ $molecule^{-1}$ $s^{-1}$, as listed in Table 1. Higher constant rates of reaction cause higher chain isomers to react with radicals more rapidly than lower chain isomers. Therefore, compared to C2-C5 isomers, the C8-C10 isomers would be removed about 2.5 times faster on average than C2 and about 40 times on average faster than C5.

The diurnal average of concentrations of PMF resolved factors appear in Figs. 4 and 5. The pattern of diurnal variations of the three ONG species factors is consistent with their degradation rate of reaction so that more reactive species cause more variant diurnal profiles of the mixing ratios than less reactive species. In the wintertime, ONG short-lived species and fuel evaporation decreased because of the effects of lower temperature; the ONG long-lived species factor, however, underwent only a moderate change. The summertime diurnal patterns of these three alkane-based factors could also be explained by the degradation rate of alkanes, except for ONG long-lived species as with longer lifetimes physical processes such as transport and dilution would have a greater impact on their ambient mixing ratios. While these factors include tracers, which are less reactive to OH, i.e., ethane and propane, they also showed lower concentrations during the daytime hours (Fig. 4). A potential reason for this decrease could be the contribution of alkane reactions with chlorine (Cl), which are known to become important under specific ambient conditions (e.g., Baker et al., 2011). Earlier studies in Houston found high Cl mixing ratios (up to 2 ppbv) under daytime sea breeze conditions (Rappenglück et al., 2011).

The speciation profile of factor 4 (Fig. 3) exhibits the percentages of a number of aromatics, including toluene (41.5% and 34.9%), *o*-xylene (40.5% and 46%), *m.p*-xylene (39.2% and 32.4%), and ethylbenzene (33.1% and 41%) for the summertime and the wintertime. The high percentages of aromatics suggest that this factor depicts the impacts of solvents from industrial and painting activities (Li et al., 2019). A similar resolved factor, shaped by aromatic compounds for summertime, also contains high percentages of alkene hydrocarbons, which include trans-2-butene (45.7%), cis-2-butene (52.5%), trans-2-pentene (%53.6), and cis-2-pentene (57.4%). Alkenes are typically associated

with incomplete fuel combustion in urban areas (Koppmann, 2008). The high percentage of 1,3-butadiene (61.8%) is most likely due to industrial releases (Czader and Rappenglueck, 2015).

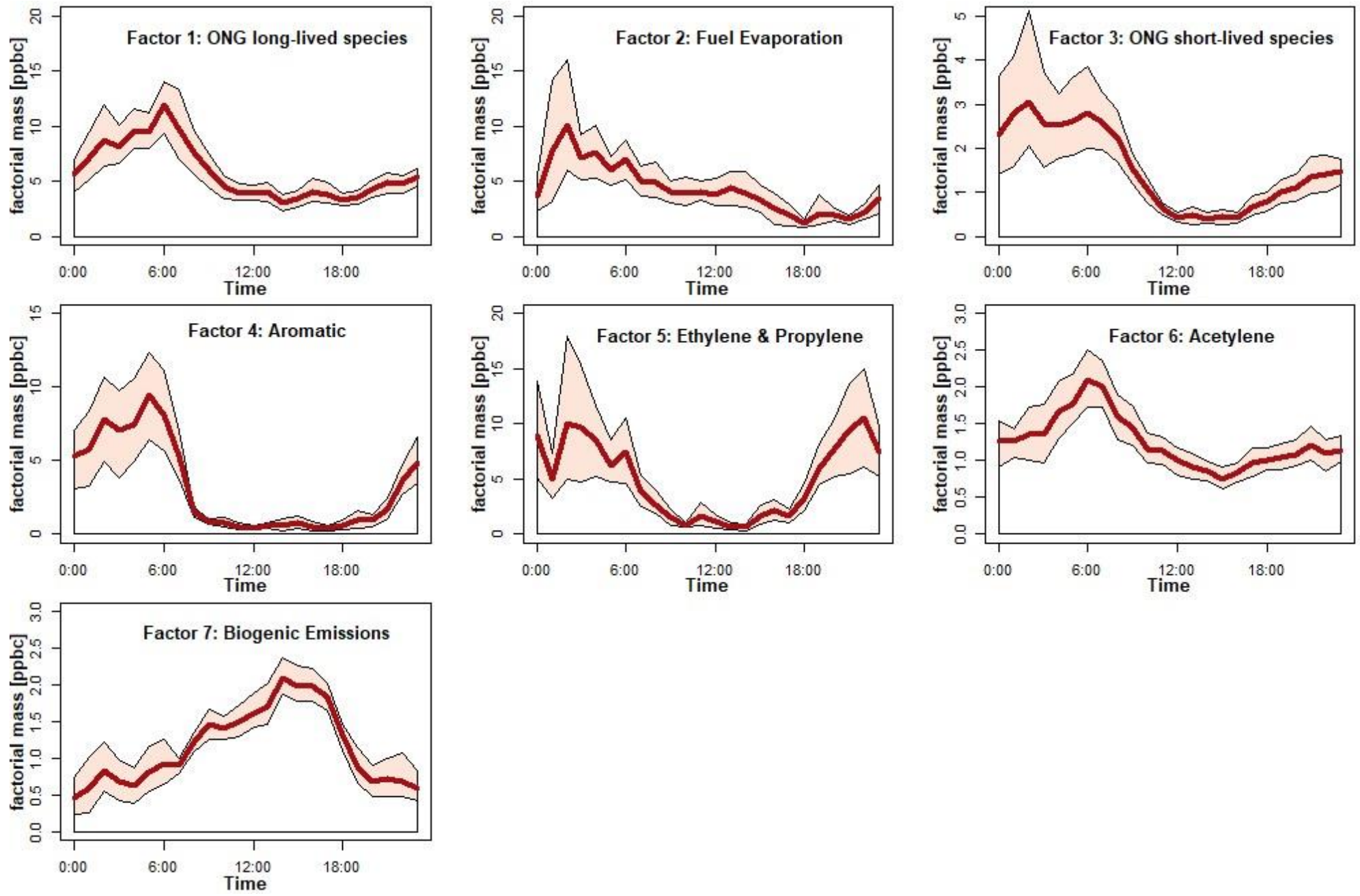

**Figure 4.** Diurnal variations of the PMF-resolved factors for the summertime at the Lynchburg Ferry station. The time is given CST (Central Standard Time). Lines correspond to the hourly average, and shaded areas present the 95% confidence intervals of the mean.

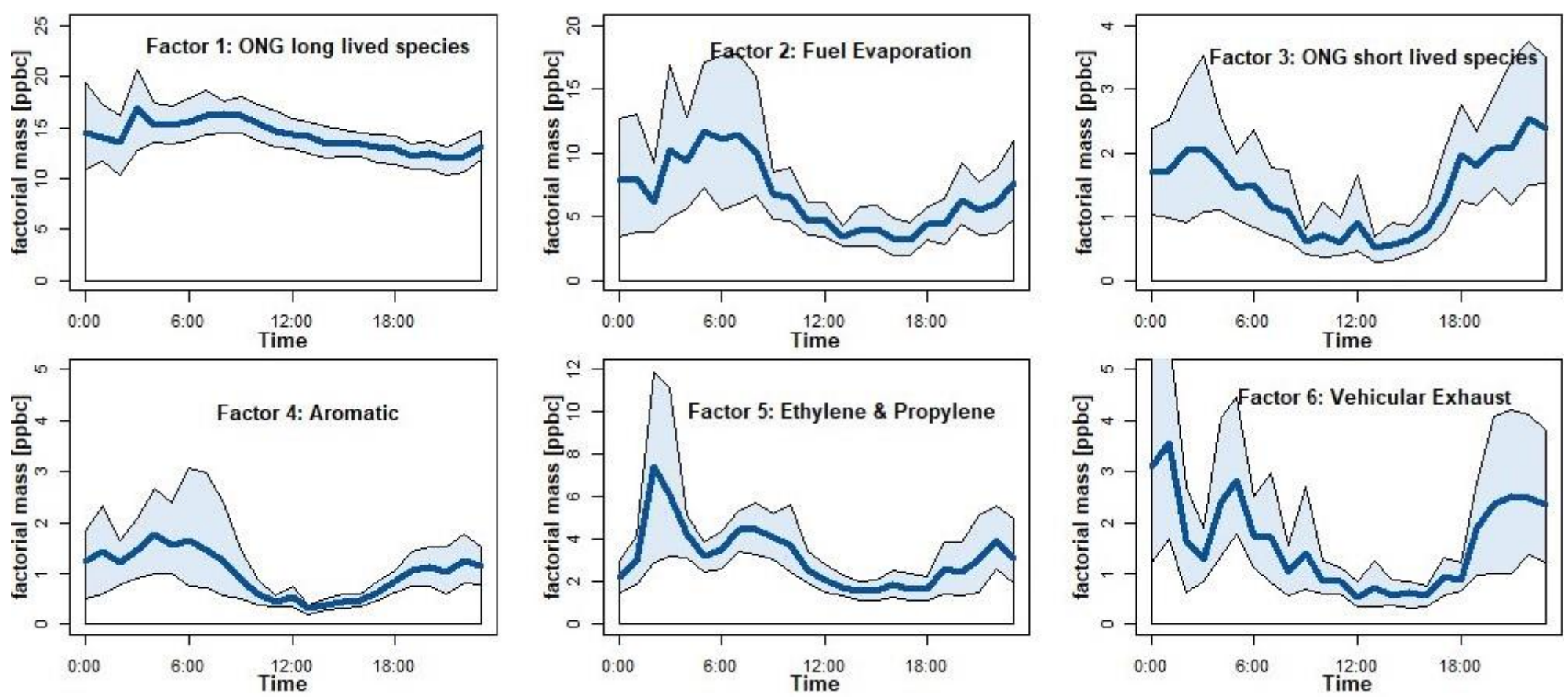

**Figure 5.** Diurnal variations of the PMF-resolved factors for the wintertime at the Lynchburg Ferry station. The time is given CST (Central Standard Time). Lines correspond to the hourly average, and shaded areas present the 95% confidence intervals of the mean.

Factor 5 of the summertime was dominated by two compounds: ethylene (82.1%) and propylene (90.8%) (Fig. 3). The corresponding percentages of ethylene and propylene in the wintertime were 80.9% and 80.4%, respectively. Ethylene is the first and propylene the second most abundant industrially produced organic compound worldwide. These two compounds are usually produced industrially by cracking petroleum hydrocarbons, and because of their double bond, they are versatile chemical feedbacks for industrial reactions (Rhew et al., 2017). Ethylene and propylene are used extensively in industrial areas, including the Houston area, as found in an earlier study (Leuchner and Rappenglueck, 2010). Based on concentration-weighted reactivities, these two alkanes are the most important VOCs in the Houston area (Czader et al., 2008). Very low concentrations of ethylene and propylene in summertime around midday are likely associated with the degradation of these very reactive compounds.

Although the factors of one to five showed similar characteristics in both the summertime and the wintertime, factors 6 and seven for the summertime and factor 6 for the wintertime presented different features. Summertime factor 6 was exclusively shaped by the highest percentage of acetylene (90%). Acetylene is known as a tracer for the combustion processes of fossil fuels, agricultural and domestic burning, and wildfires (Nicewonger et al., 2020). Acetylene measured at the Lynchburg Ferry site could have originated from industries in urban areas. Because of the low value of the OH reaction rate constant of acetylene ($0.815 \times 10^{-12}$ $cm^3$ $molecule^{-1}$ $s^{-1}$), acetylene was less variant against the higher OH concentrations in the summertime. Although mainly alkene compounds formed the sixth factor in the wintertime, a few aromatic compounds were observed in this factor. The high abundance of trans-2-butene (41.9%), cis-2-butene (52.5%), trans-2-pentene (40.3%), cis-2-pentene (38.3%), and 1,3-butadiene (64.9) could indicate the origin of vehicular exhaust in this factor. A quantitative comparison between the PMF-resolved factors for the summertime and wintertime showed an additional summertime factor that was noticeably comprised of isoprene (81.4%; Fig. 3). It is well-known that deciduous trees emit isoprene, with emission rates critically depending on solar radiation and temperature (Guenther et al., 1994; Sanadze, 2004; Sharkey et al., 2008; Sharkey and Singsaas, 1995).

## 3. 3. The source signature of ONG and photochemical processes

### **3.3.1** Source signature of emitted organic compounds

Ratios of VOCs provide helpful insights into the characterization and photochemical processes of emission sources (Wilde et al., 2021). Due to their similar OH reactivity of *iso*butane and *n*-butane, their ratio is independent of air mass dilution and mixing and can indicate what sources they have been presumably emitted from, as long as not different emission sources overlap and reactions with Cl are negligible. As seen in Section 3.3, emissions of butane isomers could be associated with natural gas, LPG, vehicular emissions, and biomass burning (Zheng et al., 2018). The ratios of *iso*butane to *n*-butane usually vary according to the specific source: 0.2-0.3 for vehicular exhaust emissions, 0.4-0.5 for LPG, and higher than 0.6 for natural gas (Buzcu and Fraser, 2006; Russo et al., 2010). Figure 6 shows that this ratio is equal to the slope of a linear two-sided fit of a correlation plot. The ratios of *iso*-butane to *n*-butane were within the range of reported emissions from LPG (0.51 in the summertime and 0.44 in the wintertime, respectively), which was similar to the results of PMF in Fig. 3. *Iso*-pentane and *n*-pentane, two other positional isomers, have similar physical and chemical features of reactivity with OH radicals, and their ratios can provide additional information about the source signature of their emissions (Gilman et al., 2013) provided that reactions with $NO_3$ are negligible (Stutz et

al., 2010). Studies have reported that *iso*-pentane to *n*-pentane ratios of 0.8 to 0.9, ~2.2 to 3.8, 1.5 to 3.0, and 1.8 to 4.6 could correspond to the origins of NG, vehicular exhausts, liquid gasoline, and fuel evaporations, respectively (Gilman et al., 2013; McGaughey et al., 2004; Watson et al., 2001). An *iso*-pentane to *n*-pentane slope of 1.3 (summertime) and 1.6 (wintertime) in this study suggested that the measured pentane tracers likely stemmed from mixed sources of fuel evaporation and emissions from natural gas (Fig. 6). This assumption is consistent with the high percentages of both compounds of *iso*-pentane and *n*-pentane in the source profile of factor 2, fuel evaporation (Fig. 3).

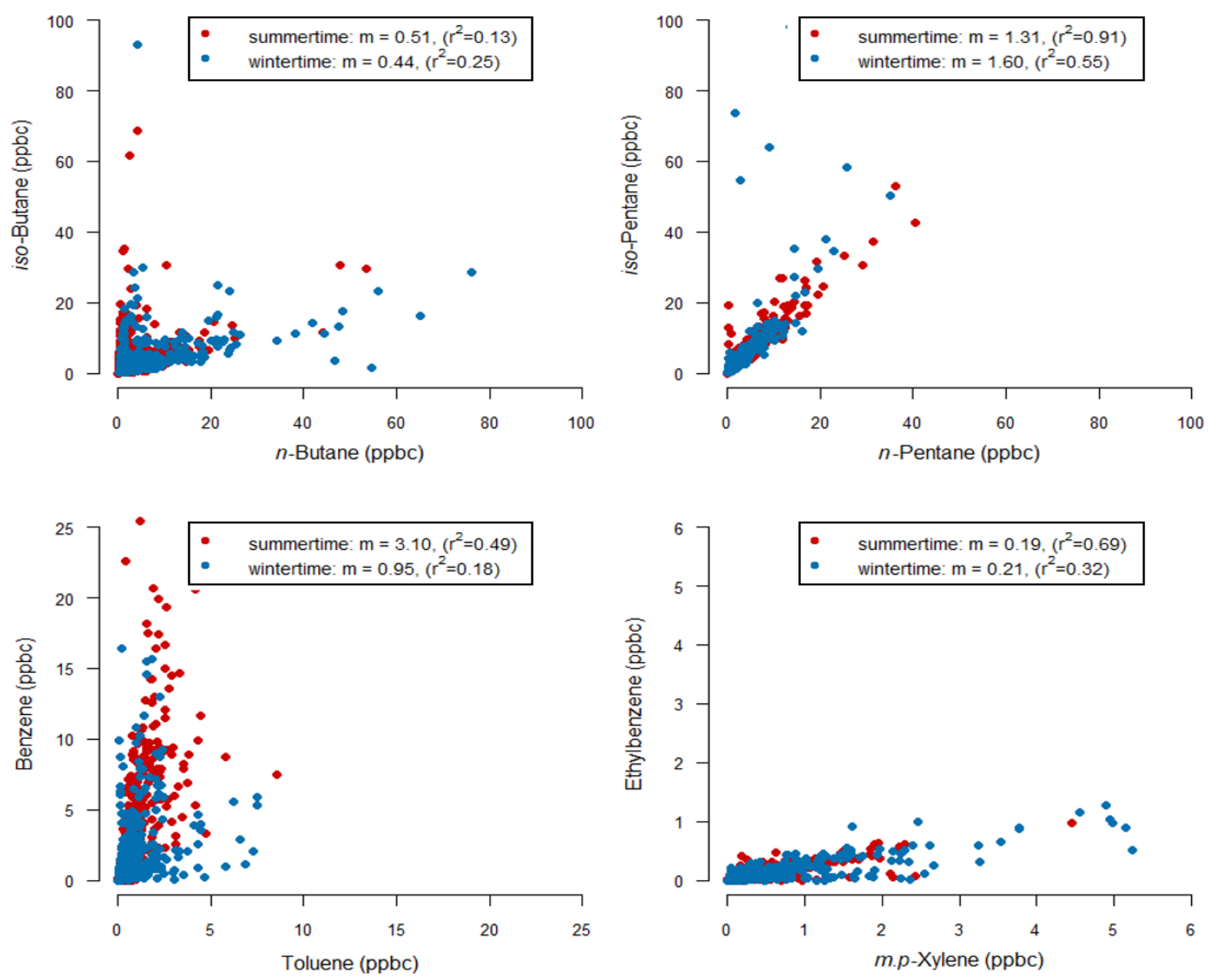

**Figure 6.** Correlation ($m$ = slope ($r^2$)) between tracers with similar reaction rate constants in the summertime and the wintertime; *iso*-butane/*n*-butane (a) and *iso*-pentane/*n*-pentane (b), and tracers with different reaction rate constants in the summertime and the wintertime; benzene/toluene (c) and ethylbenzene/*m.p*-xylene (d).

### 3.3.2. Photochemical reaction of organic compounds in the Ship Channel and the effects of VOCs on ozone formation

In addition to source signatures, the ratios of organic tracers can indicate the photochemistry of VOC compounds. Benzene, toluene, ethylbenzene, and xylenes (BTEX) are among the organic compounds known to play a significant role in the chemistry of the atmosphere (Wallace et al., 2018). BTEX is a unique group, as they exclusively react with OH, albeit at quite different reaction rates. For example, the lifetimes of the aromatic species of benzene (9.4 days), toluene (1.9 days), and ethylbenzene (1.6 days) are significantly longer than those of tracers of *o*-xylene (20.3 h) and *m.p*-xylene (19.4 h). Ratios of BTEX have been frequently used in the past to describe photochemical processes in

urban areas (Rappenglueck et al., 1998; Rappenglueck and Fabian, 1999; Winkler et al., 2002) and to estimate OH concentrations as well as the primary OH production rate (Rappenglueck et al., 2000). Consequently, we assumed that the ratios of benzene to toluene and ethylbenzene to *m.p*-xylene would be higher in the summertime. Indeed, Figure 6 supports our assumption of the seasonal variation of benzene to toluene. The figure shows the general photochemical behavior in the Ship Channel. The ratio of benzene to toluene notably increased from the wintertime (0.95) to the summertime (3.10). The ethylbenzene/*m.p*-xylene ratio, however, changed only slightly between the two periods, suggesting that the impacts of source emissions or downwind air plumes affected seasonal variation. The analysis of the nighttime data of the *m.p*-xylene to ethylbenzene ratio (4.3-4.9) show some proximity to ranges reported for near crude oil refineries (Baltrėnas et al., 2011).

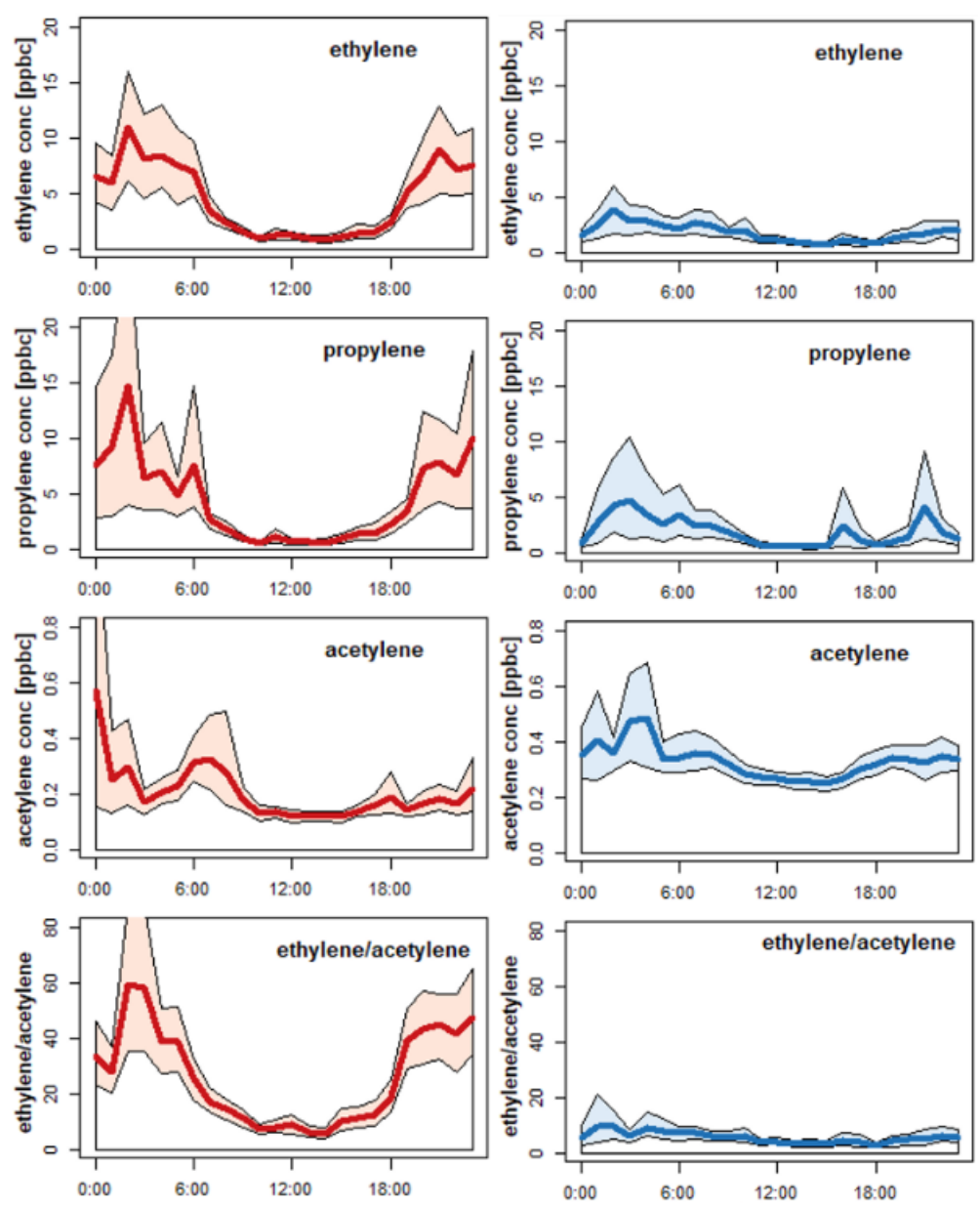

**Figure 7.** Ratios of the average diurnal values of ethylene, propylene, acetylene, and ethylene/acetylene at the Lynchburg Ferry site during the summertime (left) and the wintertime (right) 2018.

In a further investigation of the photochemistry of ethylene and propylene, we examined ratios of the measured ethylene and propylene to acetylene concentrations. Depending on their emission strengths, ethylene and propylene, two highly reactive species with considerable reaction rates with OH and $O_3$, influence the atmospheric chemistry of industrial areas. As acetylene, however, has a lower rate of reaction, the ratio of the diurnal concentrations of ethylene to acetylene could explain their influences on photochemical reaction processes. Figure 7 shows the average diurnal concentrations of ethylene, propylene, acetylene, and ethylene to acetylene at the Lynchburg Ferry site. Although the values of ethylene and propylene increased during the nighttime, they decreased during the daytime in the summer. Acetylene underwent a minor change during the day, which could partially be explained by the lower reaction rate. A

similar constant pattern existed during the wintertime, regardless of higher wintertime concentrations, which could have been the result of the lower mixing height during the colder season. A comparison between the summertime and wintertime ratios of ethylene/acetylene showed that less change in the wintertime ratios occurred, partially due to the lower temperatures and less solar radiation, critical factors that reduce the overall impacts of chemical degradation processes.

As discussed earlier, different VOCs react differently with respect to the formation of ozone in the troposphere. The ozone formation potential could be evaluated as the product of the concentration of VOCs and the MIR coefficient for MIR weighted concentration and their OH reaction rates to propylene OH reaction rate for propylene-equivalent weighted concentration, respectively. Thus, we calculated the propylene-equivalent weighted concentration and the MIR weighted concentration of 41 VOCs to investigate the impacts of individual species in the summer and winter seasons (Table S2). Calculations were based on the mean mixing ratios listed in Table 1., which would also better reflect the occurrence of transient high values relevant to ozone exceedance events compared to the median data. Among the top 10 reactive species, seven species were the same between these two methods in both seasons, differing only in the order of their rank. In the summertime, these top 10 species, as shown in Table 2, together accounted for 87.3% and 90.4% of the total ozone formation potential using propylene-equivalent and MIR methods. In the wintertime, the top reactive species of propylene-equivalent weighted concentration and method methods together accounted for 76.6% and 85.5% of total ozone formation potential in propylene-equivalent and MIR methods, respectively.

**Table 2.** Relative contributions to ozone formation by the top VOCs based on the Propyl-Equiv and MIR for summer and winter seasons 2018 at Lynchburg Ferry site

| OH Reactivity Rank | | | | MIR Rank | | | |
|---|---|---|---|---|---|---|---|
| summer | | winter | | summer | | Winter | |
| Compound | Percentage (%) | Compound | Percentage (%) | Compound | Percentage (%) | Compound | Percentage (%) |
| Propylene | 40.19 | Propylene | 29.38 | Propylene | 40.68 | Propylene | 30.72 |
| Isoprene | 18.04 | Ethylene | 9.28 | Ethylene | 32.89 | Ethylene | 23.11 |
| Ethylene | 13.64 | Styrene | 8.26 | Isoprene | 4.20 | *n*-Butane | 6.11 |
| *iso*-Pentane | 3.32 | *n*-Butane | 5.14 | *iso*-pentane | 2.90 | Propane | 5.20 |
| 1.3-Butadiene | 2.59 | 1.3-Butadiene | 4.88 | *n*-Butane | 2.00 | Ethane | 4.84 |
| *n*-Hexane | 2.21 | Propane | 4.68 | *iso*-Butane | 1.99 | *iso*-pentane | 4.11 |
| *n*-Pentane | 2.20 | *iso*-Pentane | 4.56 | *n*-Pentane | 1.78 | *iso*-butane | 3.86 |
| *n*-Butane | 1.74 | Isoprene | 4.15 | Propane | 1.39 | *n*-Pentane | 2.54 |
| *m.p*-Xylene | 1.73 | *m.p*-Xylene | 3.24 | Toluene | 1.28 | Toluene | 2.53 |
| *cis*-2-Butene | 1.62 | *n*-Pentane | 3.04 | *m.p*-Xylene | 1.26 | *m.p*-Xylene | 2.44 |

The results of Table 2 also show that six compounds out of the top 10 potential species to the ozone formation are similar in summer and winter. These six species include propylene, ethylene, iso-pentane, n-pentane, n-butane, and m.p-xylene that together contribute to (62.8% and 52.2%) of summertime ozone formation and (54.6% and 67.8%) of wintertime ozone formation using propylene-equivalent and MIT methods, respectively. Propylene and ethylene showed the highest contributions to the formation of ozone in summer and winter. This result acknowledges the earlier findings (Buzcu and Fraser, 2006; Leuchner and Rappenglueck, 2010; Zhao et al., 2004) that these two emission

factors continue to be significant emissions of VOCs, and stringent control on their emissions would be a helpful step into the mitigation of the ozone of Ship Channel.

### 3. 4. Use of backward trajectory to identify the geographic origins of VOCs over the Ship Channel

To investigate the origins of VOCs in the region, we applied the CWT method, a widely used receptor-based model that spatially reflects the concentration levels of trajectories and explores the potential geographic origins of identified source locations (Pouyaei et al., 2020; Stein et al., 2015). Since the receptor location was inside the Ship Channel, an area responsible for substantial air pollution in the region, we focused on studying local sources rather than on regional transport (Fig. 8).

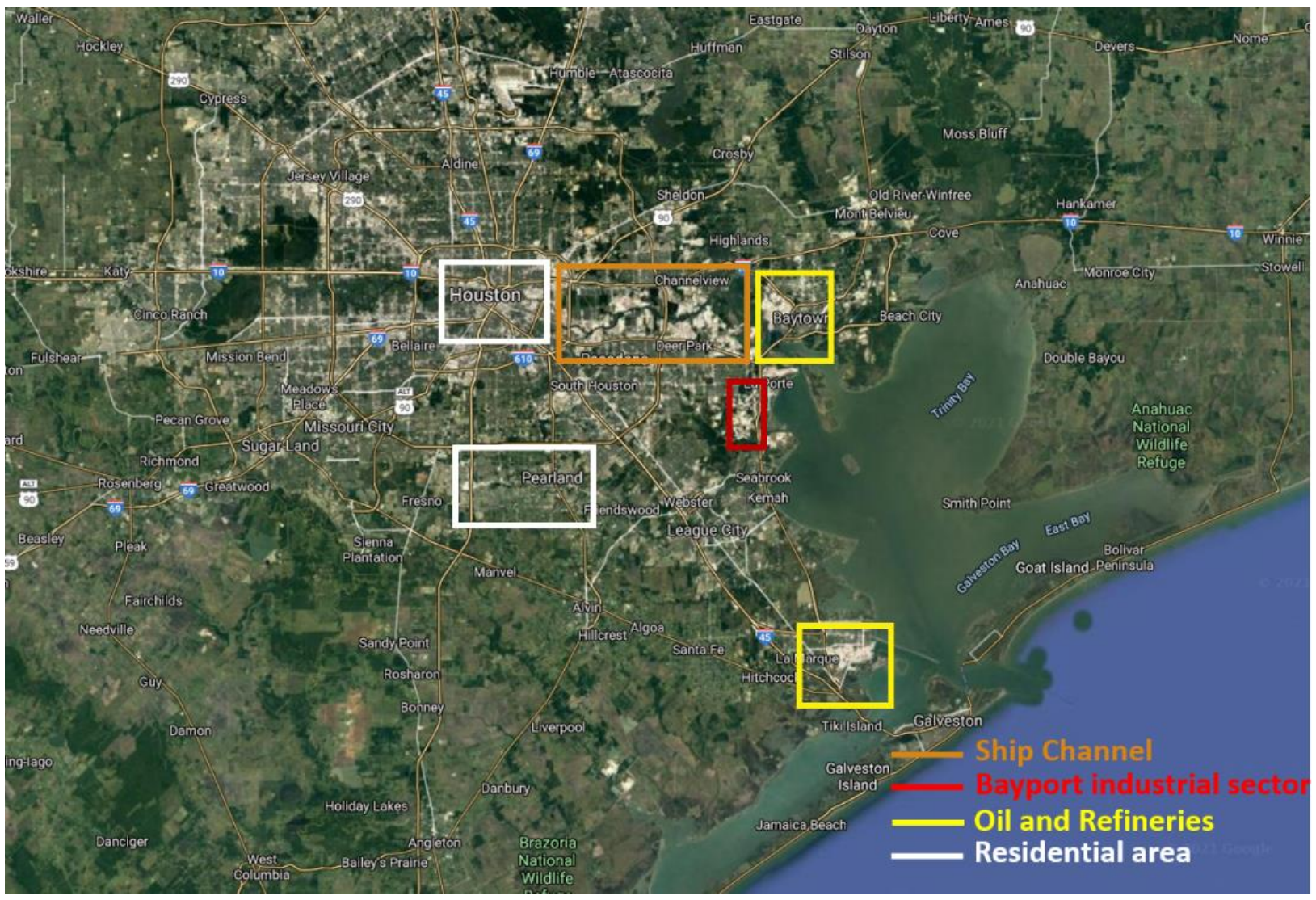

**Figure 8.** The locations of major industrial areas, oil and gas refineries, and selected residential areas in the Houston area (from © Google Maps).

During summertime 2018, back-trajectory plots revealed that the dominant cluster of trajectories originated south and southeast of the receptor location (Fig. 9). CWT results also showed that the majority of highly weighted concentration values covered the southern regions of the receptor. Upon further investigation, however, we found evidence of specific source locations in the study region. The CWT provided spatial information on the influences of VOC emissions from various industrial sectors and residential areas on the receptor site in the Ship Channel. For factor one, ONG long-lived species, we identified ship traffic and fuel tanks in the area as primary source locations (Fig. 9). Factors 2 and 3, fuel evaporation and ONG short-lived species, albeit less vivid, showed a similar distribution pattern, which could be attributed to the higher reactivity of their dominant tracers with lower contributions to total hydrocarbon emissions from refineries. Two of the world's largest refineries reside in the Houston-Galveston-Brazoria (HGB) region: The Baytown refinery northeast of the receptor site and the Galveston Bay refinery. Interestingly, CWT results for the aromatics factor showed high concentrations in an industrial area south of the receptor, the Bayport industrial district, occupied by chemical plants and industries associated with volatile chemical products (VCPs). CWT

results for factor 5, ethylene and propylene, pointed toward industrial complexes mixed with highways and residential areas in the region. Factor 6, acetylene, showed higher concentrations near the Baytown industrial refinery. The CWT results for this factor cover a broader area south of the station, which we could attribute to the long lifetime of acetylene. In addition, high CWT values indicated forests/parks in the Pearland and Brazoria region as sources of biogenic emissions. Investigation of the daytime and nighttime results of CWT confirmed the impacts of industrial and refinery sectors around the southern region of the study domain, especially in summertime (Fig. S2). In addition, the vivid presence of daytime biogenic emissions is correlated with the higher daytime temperature.

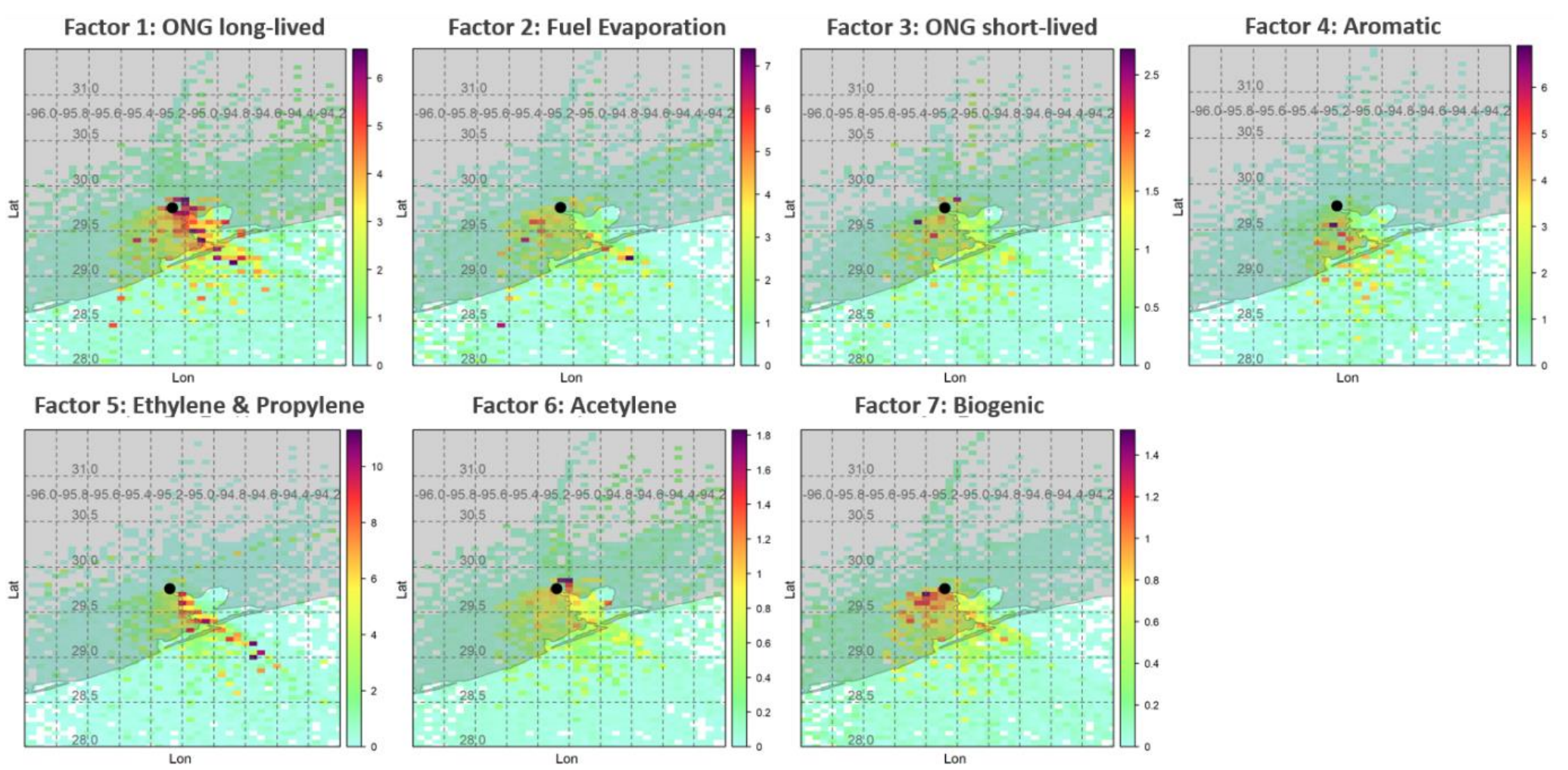

**Figure 9.** Seasonal concentration-weighted trajectories for seven sources derived from the PMF analysis for the summertime. The factors included ONG long-lived species, fuel evaporation, ONG short-lived species, aromatic, ethylene and propylene, acetylene, and biogenic emissions. The black dot represents the Lynchburg Ferry site.

During wintertime 2018, the trajectory paths emerged from several directions around the Lynchburg Ferry. The results indicate the emission sources of VOCs are influenced by the plumes of air masses near Ship Channel (Fig. 10). A relatively higher level of concentrations over northern areas could be associated with a lower atmospheric boundary layer in the wintertime which covers a majority of the domain. The wintertime results of CWT showed that the spatial distribution of the first factor, ONG long-lived species, in the winter was similar to that in the summertime. Potential origins of this factor were the Baytown refinery and emissions from areas to the northeast, home to a number of fuel tanks filled with ethane and propane. Factors 2 and 3, fuel evaporation and ONG short-lived species, mainly originated from the Galveston Bay refinery. Wintertime aromatic compounds appear to be affected by the Bayport industrial segments and, to a lesser extent, from the downtown area. The origin of wintertime ethylene and propylene can be attributed to various highways and residential areas, subject to different wind directions. In the winter season, results of the vehicular exhaust factor pointed to highways and residential regions with a substantial number of vehicles, as well as to ship traffic in the Ship Channel area. CWT results of each factor for daytime and nighttime were shown in Fig. S3.

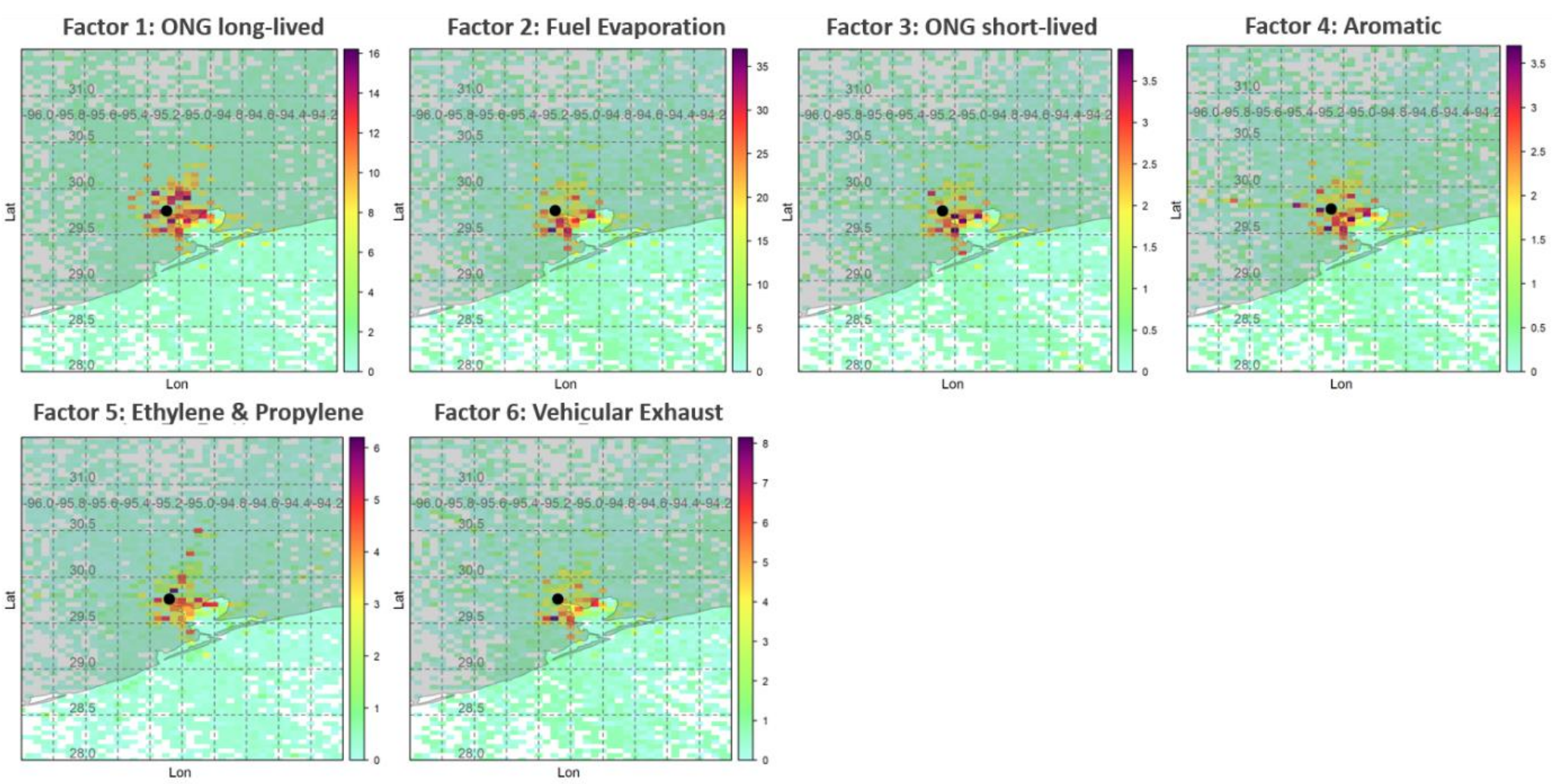

**Figure 10.** Seasonal concentration-weighted trajectory for six identified sources derived from the PMF analysis in the wintertime. The factors include ONG long-lived species, fuel evaporation, ONG short-lived species, aromatic, ethylene and propylene, vehicular exhaust, and biogenic emissions. The black dot represents the Lynchburg Ferry site.

An overall comparison of the CWT results for the two seasons showed similar potential sources for most of the distributed VOCs. The geography of source emissions for some factors (i.e., ethylene and propylene, and vehicular exhaust) might have varied due to seasonal changes in terms of the wind direction. The ONG long-lived species factor in both seasons indicated both Baytown and Galveston Bay refineries as potential origins; the potential source of the ONG short-lived species in each season, however, differed. The results of factor 2 of fuel evaporation, factor 3 of ONG short-lived species, factor 4 of aromatic, and factor 5 of ethylene and propylene show a similar spatial distribution of the source emissions between two seasons, although the seasonal discrepancies could be due to different wind patterns, affecting the transport of the VOCs over the Ship Channel.

**4-Summary**

For two sampling periods, summertime and wintertime of 2018, we investigated the characteristics of the VOCs over the Houston Ship Channel using receptor modeling and the CWT dispersion model. We evaluated the levels and compositions of VOC compounds, identified their emission sources, and identified their possible origins. Our results found that average VOCs concentrations were 28.68 ppbC in the summertime and 33.92 ppbC in the wintertime at the Lynchburg Ferry site. In both seasons, the dominant groups of VOCs were the alkane compounds which contributed to the 61% and 83% of total VOC concentrations in summertime and wintertime, respectively. Studying the source profiles and diurnal variations of the PMF-resolved factors identified three types of ONG emission factors, which differ in their reaction rates. The source apportionment identified two other factors, —aromatics, along with ethylene and propylene— that were present in both summertime and wintertime. Acetylene and biogenic emission factors were found to be present for the summertime, and vehicular exhaust was shown to be effective for the wintertime. In terms of reactivity-based concentration of VOCs, alkene had the highest ozone formation potential in summertime propylene

(40.2% and 40.7%) and ethylene (13.6% and 32.9%) and wintertime propylene (29.4% and 30.7%), and ethylene (9.3% and 23.1%) through the kinetic and mechanism reactivity methods respectively. In addition, the summertime isoprene makes a noticeable contribution to ozone formation (18.0%) with respect to its high OH reactivity.

We also examined the geographical location origins of the emissions sources of VOCs compounds using trajectory analysis. During both seasons, the measurements of VOCs at the Lynchburg Ferry site were influenced by emissions from the petrochemical sector and industrial complexes within the Houston industrial area, especially the Baytown refineries and Bayport industrial district. The results also showed parts of the VOCs at the site originated from the Galveston Bay refineries. The CWT analysis results also indicated parts of the measured aromatic compounds were affected by the plumes of air masses transported from the south of the receptor where the Bayport industrial district is located.

The findings of this study suggest two different policy approaches: (1) targeting the alkane emissions sources regarding their higher contributions to the total organic compounds or (2) focusing on other groups of VOCs (e.g., aromatics and alkenes), which are less relevant to the energy utilization policies and are more linked to the industrial productions. Findings from prior studies for the Houston area have led to significant emission reductions in highly reactive VOCs over the last 15 years. However, the results of this study admit the importance of the ethylene and propylene emissions which continue to be an issue for the mitigation of ozone.

**Declaration of competing interest**

The authors declare no competing financial and/or non-financial interests in relation to the work described.

**Acknowledgments**

This study was partially supported by the High Priority Area Research Seed Grant of the University of Houston.

**5-References**

Atkinson, R.: Atmospheric chemistry of VOCs and NO(x), Atmos. Environ., 34(12–14), 2063–2101, doi:10.1016/S1352-2310(99)00460-4, 2000.

Atkinson, R. and Arey, J.: Atmospheric Degradation of Volatile Organic Compounds, Chem. Rev., doi:10.1021/cr0206420, 2003.

Atkinson, R., Baulch, D. L., Cox, R. A., Hampson, R. F., Kerr, J. A., Rossi, M. J. and Troe, J.: Evaluated Kinetic, Photochemical and Heterogeneous Data for Atmospheric Chemistry: Supplement V: IUPAC Subcommittee on Gas Kinetic Data Evaluation for Atmospheric Chemistry, J. Phys. Chem. Ref. Data, doi:10.1063/1.556011, 1997.

Baker, A. K., Rauthe-Schch, A., Schuck, T. J., Brenninkmeijer, C. A. M., Van Velthoven, P. F. J., Wisher, A. and Oram, D. E.: Investigation of chlorine radical chemistry in the Eyjafjallajkull volcanic plume using observed depletions in non-methane hydrocarbons, Geophys. Res. Lett., 38(13), doi:10.1029/2011GL047571, 2011.

Baltrėnas, P., Baltrėnaitė, E., Šerevičienė, V. and Pereira, P.: Atmospheric BTEX concentrations in the vicinity of the crude oil refinery of the Baltic region, Environ. Monit. Assess., 182(1–4), 115–127, doi:10.1007/s10661-010-1862-0, 2011.

Berger, B. and Anderson, K.: Modern petroleum: a basic primer of the industry, 1981.

Bi, S., Kiaghadi, A., Schulze, B. C., Bernier, C., Bedient, P. B., Padgett, J. E., Rifai, H. and Griffin, R. J.: Simulation of potential formation of atmospheric pollution from aboveground storage tank leakage after severe storms, Atmos. Environ., 248, 118225, doi:10.1016/j.atmosenv.2021.118225, 2021.

Burton and B.: Volatile Organic Compounds, Indoor Air Pollut. Heal. [online] Available from: https://ci.nii.ac.jp/naid/10018780221 (Accessed 17 June 2021), 1997.

Buzcu, B. and Fraser, M. P.: Source identification and apportionment of volatile organic compounds in Houston, TX, Atmos. Environ., 40(13), 2385–2400, doi:10.1016/j.atmosenv.2005.12.020, 2006.

Carter, W. P. L.: Development of ozone reactivity scales for volatile organic compounds, J. Air Waste Manag. Assoc., 44(7), 881–899, doi:10.1080/1073161x.1994.10467290, 1994.

Carter, W. P. L.: Development of the SAPRC-07 chemical mechanism, Atmos. Environ., 44(40), 5324–5335, doi:10.1016/j.atmosenv.2010.01.026, 2010.

Chameides, W. L., Fehsenfeld, F., Rodgers, M. O., Cardelino, C., Martinez, J., Parrish, D., Lonneman, W., Lawson, D. R., Rasmussen, R. A., Zimmerman, P., Greenberg, J., Middleton, P. and Wang, T.: Ozone precursor relationships in the ambient atmosphere, J. Geophys. Res., 97(D5), 6037–6055, doi:10.1029/91JD03014, 1992.

Czader, B. H. and Rappenglueck, B.: Modeling of 1,3-butadiene in urban and industrial areas, Atmos. Environ., 102, 30–42, doi:10.1016/j.atmosenv.2014.11.039, 2015.

Czader, B. H., Byun, D. W., Kim, S. T. and Carter, W. P. L.: A study of VOC reactivity in the Houston-Galveston air mixture utilizing an extended version of SAPRC-99 chemical mechanism, Atmos. Environ., doi:10.1016/j.atmosenv.2008.01.039, 2008.

Dalsøren, S. B., Myhre, G., Hodnebrog, O., Myhre, C. L., Stohl, A., Pisso, I., Schwietzke, S., Höglund-Isaksson, L., Helmig, D., Reimann, S., Sauvage, S., Schmidbauer, N., Read, K. A., Carpenter, L. J., Lewis, A. C., Punjabi, S. and Wallasch, M.: Discrepancy between simulated and observed ethane and propane levels explained by underestimated fossil emissions /704/106/35/824 /704/172/169/824 /119 article, Nat. Geosci., 11(3), 178–184, doi:10.1038/s41561-018-0073-0, 2018.

Diao, L., Choi, Y., Czader, B., Li, X., Pan, S., Roy, A., Souri, A. H., Estes, M. and Jeon, W.: Discrepancies between modeled and observed nocturnal isoprene in an urban environment and the possible causes: A case study in Houston, Atmos. Res., 181, 257–264, doi:10.1016/j.atmosres.2016.07.009, 2016.

Dodge, M. C.: Combined effects of organic reactivity and NMHC/NOx ratio on photochemical oxidant formation-a modeling study, Atmos. Environ., 18(8), 1657–1665, doi:10.1016/0004-6981(84)90388-3, 1984.

Duan, J., Tan, J., Yang, L., Wu, S. and Hao, J.: Concentration, sources and ozone formation potential of volatile organic compounds (VOCs) during ozone episode in Beijing, Atmos. Res., 88(1), 25–35, doi:10.1016/j.atmosres.2007.09.004, 2008.

EPA: Technical Assistance Document for Sampling and Analysis of Ozone Precursors for the Photochemical Assessment Monitoring Stations Program - Revision 2 - April 2019. [online] Available from: https://www.epa.gov/sites/production/files/2019-11/documents/pams_technical_assistance_document_revision_2_april_2019.pdf (Accessed 17 June 2021), 2019.

Forkel, R., Klemm, O., Graus, M., Rappenglueck, B., Stockwell, W. R., Grabmer, W., Held, A., Hansel, A. and Steinbrecher, R.: Trace gas exchange and gas phase chemistry in a Norway spruce forest: A study with a coupled 1-dimensional canopy atmospheric chemistry emission model, Atmos. Environ., 40, 28–42, doi:10.1016/j.atmosenv.2005.11.070, 2006.

Fuentes, J. D., Lerdau, M., Atkinson, R., Baldocchi, D., Bottenheim, J. W., Ciccioli, P., Lamb, B., Geron, C., Gu, L., Guenther, A., Sharkey, T. D. and Stockwell, W.: Biogenic Hydrocarbons in the Atmospheric Boundary Layer: A Review, Bull. Am. Meteorol. Soc., doi:10.1175/1520-0477(2000)081<1537:BHITAB>2.3.CO;2, 2000.

Gilman, J. B., Lerner, B. M., Kuster, W. C. and De Gouw, J. A.: Source signature of volatile organic compounds from oil and natural gas operations in northeastern Colorado, Environ. Sci. Technol., 47(3), 1297–1305, doi:10.1021/es304119a, 2013.

Guenther, A., Zimmerman, P. and Wildermuth, M.: Natural volatile organic compound emission rate estimates for U.S. woodland landscapes, Atmos. Environ., 28(6), 1197–1210, doi:10.1016/1352-2310(94)90297-6, 1994.

Heald, C. L., Gouw, J. De, Goldstein, A. H., Guenther, A. B., Hayes, P. L., Hu, W., Isaacman-Vanwertz, G., Jimenez, J. L., Keutsch, F. N., Koss, A. R., Misztal, P. K., Rappenglück, B., Roberts, J. M., Stevens, P. S., Washenfelder, R. A., Warneke, C. and Young, C. J.: Contrasting Reactive Organic Carbon Observations in the Southeast United States (SOAS) and Southern California (CalNex), Environ. Sci. Technol., 54(23), 14923–14935, doi:10.1021/acs.est.0c05027, 2020.

Henry, R. C. and Hidy, G. M.: Multivariate analysis of particulate sulfate and other air quality variables by principal components-Part I. Annual data from Los Angeles and New York, Atmos. Environ., 13(11), 1581–1596, doi:10.1016/0004-6981(79)90068-4, 1979.

Hopke, P. K.: Recent developments in receptor modeling, J. Chemom., 17(5), 255–265, doi:10.1002/cem.796, 2003.

Hopke, P. K.: Review of receptor modeling methods for source apportionment, J. Air Waste Manag. Assoc., 66(3), 237–259, doi:10.1080/10962247.2016.1140693, 2016.

Izumi, K. and Fukuyama, T.: Photochemical aerosol formation from aromatic hydrocarbons in the presence of NOx, Atmos. Environ. Part A, Gen. Top., 24(6), 1433–1441, doi:10.1016/0960-1686(90)90052-O, 1990.

Jobson, B. T., Berkowitz, C. M., Kuster, W. C., Goldan, P. D., Williams, E. J., Fesenfeld, F. C., Apel, E. C., Karl, T., Lonneman, W. A. and Riemer, D.: Hydrocarbon source signatures in Houston, Texas: Influence of the petrochemical industry, J. Geophys. Res. D Atmos., 109(24), 1–26, doi:10.1029/2004JD004887, 2004.

Kaiser, H. F.: A second generation little jiffy, Psychometrika, doi:10.1007/BF02291817, 1970.

Karl, T., Guenther, A., Spirig, C., Hansel, A. and Fall, R.: Seasonal variation of biogenic VOC emissions above a mixed hardwood forest in northern Michigan, Res. Lett, 30(23), 2186, doi:10.1029/2003GL018432, 2003.

Kesselmeier, J. and Staudt, M.: Biogenic volatile organic compounds (VOC): An overview on emission, physiology and ecology, J. Atmos. Chem., 33(1), 23–88, doi:10.1023/A:1006127516791, 1999.

Kim, H. Y., Lee, J. D., Kim, J. Y., Lee, J. Y., Bae, O. N., Choi, Y. K., Baek, E., Kang, S., Min, C., Seo, K., Choi, K., Lee, B. M. and Kim, K. B.: Risk assessment of volatile organic compounds (VOCs) detected in sanitary pads, J. Toxicol. Environ. Heal. - Part A Curr. Issues, 82(11), 678–695, doi:10.1080/15287394.2019.1642607, 2019.

Kleinman, L. I., Daum, P. H., Imre, D., Lee, Y. N., Nunnermacker, L. J., Springston, S. R., Weinstein-Lloyd, J. and Rudolph, J.: Ozone production rate and hydrocarbon reactivity in 5 urban areas: A cause of high ozone concentration in Houston, Geophys. Res. Lett., 29(10), 105–106, doi:10.1029/2001GL014569, 2002.

Koppmann, R.: Volatile Organic Compounds in the Atmosphere, John Wiley Sons [online] Available from:

https://books.google.com/books?hl=en&lr=&id=uUN3lFs_pgoC&oi=fnd&pg=PP2&dq=volatile+organic+chemistry+in+the+atmosphere&ots=HGf3q0HFCQ&sig=R7sUo5q9W3BV7fyJXP8K7j_XvqI#v=onepage&q=volatile organic chemistry in the atmosphere&f=false (Accessed 17 June 2021), 2008.

Kumar, A., Sinha, V., Shabin, M., Hakkim, H., Bonsang, B. and Gros, V.: Non-methane hydrocarbon (NMHC) fingerprints of major urban and agricultural emission sources for use in source apportionment studies, Atmos. Chem. Phys., 20(20), 12133–12152, doi:10.5194/acp-20-12133-2020, 2020.

Leuchner, M. and Rappenglueck, B.: VOC source-receptor relationships in Houston during TexAQS-II, , doi:10.1016/j.atmosenv.2009.02.029, 2010.

Li, M., Zhang, Q., Zheng, B., Tong, D., Lei, Y., Liu, F., Hong, C., Kang, S., Yan, L., Zhang, Y., Bo, Y., Su, H., Cheng, Y. and He, K.: Persistent growth of anthropogenic non-methane volatile organic compound (NMVOC) emissions in China during 1990-2017: Drivers, speciation and ozone formation potential, Atmos. Chem. Phys., 19(13), 8897–8913, doi:10.5194/acp-19-8897-2019, 2019.

Liu, S. C.: Ozone production in the rural troposphere and the implications for regional and global ozone distributions., J. Geophys. Res., 92(D4), 4191–4207, doi:10.1029/JD092iD04p04191, 1987.

Liu, Y., Wang, H., Jing, S., Peng, Y., Gao, Y., Yan, R., Wang, Q., Lou, S., Cheng, T. and Huang, C.: Strong regional transport of volatile organic compounds (VOCs) during wintertime in Shanghai megacity of China, Atmos. Environ., 244, 117940, doi:10.1016/j.atmosenv.2020.117940, 2021.

Logan, J. A., Prather, M. J., Wofsy, S. C. and McElroy, M. B.: Tropospheric chemistry: a global perspective., J. Geophys. Res., 86(C8), 7210–7254, doi:10.1029/JC086iC08p07210, 1981.

Maroni, M., Seifert, B. and Lindvall, T.: Indoor Air Quality: A Comprehensive Reference Book, [online] Available from: https://books.google.com/books?hl=en&lr=&id=qsyLaKnn-nYC&oi=fnd&pg=PP1&dq=Maroni,+M.,+Seifert,+B.,+Lindvall,+T.+(1995).+Indoor+air+quality:+a+comprehensive+reference+book.+Elsevier.&ots=mFtnIMwciU&sig=PECCOA9LoZkj-UcsdRHZrJsiYU0#v=onepage&q=Maroni%252C M.%252 (Accessed 17 June 2021), 1995.

Marvin, M. R., Palmer, P. I., Latter, B. G., Siddans, R., Kerridge, B. J., Talib Latif, M. and Firoz Khan, M.: Photochemical environment over Southeast Asia primed for hazardous ozone levels with influx of nitrogen oxides from seasonal biomass burning, Atmos. Chem. Phys., 21(3), 1917–1935, doi:10.5194/acp-21-1917-2021, 2021.

McGaughey, G. R., Desai, N. R., Allen, D. T., Seila, R. L., Lonneman, W. A., Fraser, M. P., Harley, R. A., Pollack, A. K., Ivy, J. M. and Price, J. H.: Analysis of motor vehicle emissions in a Houston tunnel during the Texas Air Quality Study 2000, Atmos. Environ., 38(20), 3363–3372, doi:10.1016/j.atmosenv.2004.03.006, 2004.

Mølhave, L.: Volatile Organic Compounds, Indoor Air Quality and Health, Indoor Air, 1(4), 357–376, doi:10.1111/j.1600-0668.1991.00001.x, 1991.

Mousavinezhad, S., Choi, Y., Pouyaei, A., Ghahremanloo, M. and Nelson, D. L.: A comprehensive investigation of surface ozone pollution in China, 2015–2019: Separating the contributions from meteorology and precursor emissions, Atmos. Res., 257, 105599, doi:10.1016/j.atmosres.2021.105599, 2021.

Nault, B., Jo, D., McDonald, B., Campuzano-Jost, P., Day, D., Hu, W., Schroder, J., Allan, J., Blake, D., Canagaratna, M., Coe, H., Coggon, M., DeCarlo, P., Diskin, G., Dunmore, R., Flocke, F., Fried, A., Gilman, J., Gkatzelis, G., Hamilton, J., Hanisco, T., Hayes, P., Henze, D., Hodzic, A., Hopkins, J., Hu, M., Huey, L. G., Jobson, B. T., Kuster, W., Lewis, A., Li, M., Liao, J., Nawaz, M. O., Pollack, I., Peischl, J., Rappenglück, B., Reeves, C., Richter, D., Roberts, J., Ryerson, T., Shao, M., Sommers, J., Walega, J., Warneke, C., Weibring, P., Wolfe, G., Young, D., Yuan, B., Zhang, Q., de Gouw, J. and Jimenez, J.: Anthropogenic Secondary Organic Aerosols Contribute Substantially to Air Pollution Mortality, Atmos. Chem. Phys., 1–53, doi:10.5194/acp-2020-914, 2020.

Ng, N. L., Kroll, J. H., Chan, A. W. H., Chhabra, P. S., Flagan, R. C. and Seinfeld, J. H.: Secondary organic aerosol formation from m-xylene, toluene, and benzene, Atmos. Chem. Phys., 7(14), 3909–3922, doi:10.5194/acp-7-3909-2007, 2007.

Nicewonger, M. R., Aydin, M., Prather, M. J. and Saltzman, E. S.: Reconstruction of Paleofire Emissions Over the Past Millennium From Measurements of Ice Core Acetylene, Geophys. Res. Lett., 47(3), e2019GL085101, doi:10.1029/2019GL085101, 2020.

Norris, G., Duvall, R., Brown, S. and Bai, S.: EPA Positive Matrix Factorization (PMF) 5.0 Fundamentals and User Guide., 2014.

Odum, J. R., Jungkamp, T. P. W., Griffin, R. J., Forstner, H. J. L., Flagan, R. C. and Seinfeld, J. H.: Aromatics, reformulated gasoline, and atmospheric organic aerosol formation, Environ. Sci. Technol., 31(7), 1890–1897, doi:10.1021/es960535l, 1997.

Paatero, P.: The Multilinear Engine—A Table-Driven, Least Squares Program for Solving Multilinear Problems, Including the n-Way Parallel Factor Analysis Model, J. Comput. Graph. Stat., 8(4), 854–888, doi:10.1080/10618600.1999.10474853, 1999.

Paatero, P. and Tapper, U.: Positive matrix factorization: A non-negative factor model with optimal utilization of error estimates of data values, Environmetrics, 5(2), 111–126, doi:10.1002/env.3170050203, 1994.

Pan, S., Choi, Y., Roy, A., Li, X., Jeon, W. and Souri, A. H.: Modeling the uncertainty of several VOC and its impact on simulated VOC and ozone in Houston, Texas, Atmos. Environ., 120, 404–416, doi:10.1016/j.atmosenv.2015.09.029, 2015.

Piccot, S. D., Watson, J. J. and Jones, J. W.: A global inventory of volatile organic compound emissions from anthropogenic sources, J. Geophys. Res., 97(D9), 9897–9912, doi:10.1029/92JD00682, 1992.

Port Houston: Overview - Port Houston, [online] Available from: https://porthouston.com/about-us/ (Accessed 17 June 2021), 2019.

Pouyaei, A., Choi, Y., Jung, J., Sadeghi, B. and Han Song, C.: Concentration Trajectory Route of Air pollution with an Integrated Lagrangian model (C-TRAIL Model v1.0) derived from the Community Multiscale Air Quality Model (CMAQ Model v5.2), Geosci. Model Dev., 13(8), 3489–3505, doi:10.5194/gmd-13-3489-2020, 2020.

Rappenglück, B., Lefer, B., Mellqvist, J., Czader, B., Golovko, J., Li, X., Alvarez, S., Haman, C. and Johansson, J.: University of Houston Study of Houston Atmospheric Radical Precursors (SHARP), Rep. to Texas Comm. Environ. Qual. Texas Comm. Environ. Qual. Austin, Texas, USA, 2011.

Rappenglueck, B. and Fabian, P.: Nonmethane hydrocarbons (NMHC) in the Greater Munich Area/Germany, Atmos. Environ., 33(23), 3843–3857, doi:10.1016/S1352-2310(98)00394-X, 1999.

Rappenglueck, B., Fabian, P., Kalabokas, P., Viras, L. G. and Ziomas, I. C.: Quasi-continuous measurements of non-methane hydrocarbons (NMHC) in the greater Athens area during MEDCAPHOT-TRACE, in Atmospheric Environment, vol. 32, pp. 2103–2121, Pergamon., 1998.

Rappenglueck, B., Oyola, P., Olaeta, I. and Fabian, P.: The evolution of photochemical smog in the Metropolitan Area of Santiago de Chile, J. Appl. Meteorol., doi:10.1175/1520-0450(2000)039<0275:TEOPSI>2.0.CO;2, 2000.

Rappenglueck, B., Schmitz, R., Bauerfeind, M., Cereceda-Balic, F., Von Baer, D., Jorquera, H., Silva, Y. and Oyola, P.: An urban photochemistry study in Santiago de Chile, Atmos. Environ., 39(16), 2913–2931, doi:10.1016/j.atmosenv.2004.12.049, 2005.

Rhew, R. C., Deventer, M. J., Turnipseed, A. A., Warneke, C., Ortega, J., Shen, S., Martinez, L., Koss, A., Lerner,

B. M., Gilman, J. B., Smith, J. N., Guenther, A. B. and De Gouw, J. A.: Ethene, propene, butene and isoprene emissions from a ponderosa pine forest measured by relaxed eddy accumulation, Atmos. Chem. Phys., 17(21), 13417–13438, doi:10.5194/acp-17-13417-2017, 2017.

Russo, R. S., Zhou, Y., White, M. L., Mao, H., Talbot, R. and Sive, B. C.: Multi-year (2004-2008) record of nonmethane hydrocarbons and halocarbons in New England: Seasonal variations and regional sources, Atmos. Chem. Phys., 10(10), 4909–4929, doi:10.5194/acp-10-4909-2010, 2010.

Sadeghi, B., Choi, Y., Yoon, S., Flynn, J., Kotsakis, A. and Lee, S.: The characterization of fine particulate matter downwind of Houston: Using integrated factor analysis to identify anthropogenic and natural sources, Environ. Pollut., 262, 114345, doi:10.1016/j.envpol.2020.114345, 2020.

Sanadze, G. A.: Biogenic isoprene (a review), Russ. J. Plant Physiol., 51(6), 729–741, doi:10.1023/B:RUPP.0000047821.63354.a4, 2004.

Sharkey, T. D. and Singsaas, E. L.: Why plants emit isoprene, Nature, 374(6525), 769, doi:10.1038/374769a0, 1995.

Sharkey, T. D., Wiberley, A. E. and Donohue, A. R.: Isoprene emission from plants: Why and how, Ann. Bot., doi:10.1093/aob/mcm240, 2008.

Song, S. K., Choi, Y. N., Choi, Y., Flynn, J. and Sadeghi, B.: Characteristics of aerosol chemical components and their impacts on direct radiative forcing at urban and suburban locations in Southeast Texas, Atmos. Environ., 246, 118151, doi:10.1016/j.atmosenv.2020.118151, 2021.

Stein, A. F., Draxler, R. R., Rolph, G. D., Stunder, B. J. B., Cohen, M. D. and Ngan, F.: Noaa's hysplit atmospheric transport and dispersion modeling system, Bull. Am. Meteorol. Soc., doi:10.1175/BAMS-D-14-00110.1, 2015.

Stutz, J., Wong, K. W., Lawrence, L., Ziemba, L., Flynn, J. H., Rappenglueck, B. and Lefer, B.: Nocturnal NO3 radical chemistry in Houston, TX, Atmos. Environ., doi:10.1016/j.atmosenv.2009.03.004, 2010.

Tsigaridis, K. and Kanakidou, M.: Global modelling of secondary organic aerosol in the troposphere: A sensitivity analysis, Atmos. Chem. Phys., 3(5), 1849–1869, doi:10.5194/acp-3-1849-2003, 2003.

Wallace, H. W., Sanchez, N. P., Flynn, J. H., Erickson, M. H., Lefer, B. L. and Griffin, R. J.: Source apportionment of particulate matter and trace gases near a major refinery near the Houston Ship Channel, Atmos. Environ., 173, 16–29, doi:10.1016/j.atmosenv.2017.10.049, 2018.

Warneke, C., Geiger, F., Edwards, P. M., Dube, W., Pétron, G., Kofler, J., Zahn, A., Brown, S. S., Graus, M., Gilman, J. B., Lerner, B. M., Peischl, J., Ryerson, T. B., De Gouw, J. A. and Roberts, J. M.: Volatile organic compound emissions from the oil and natural gas industry in the Uintah Basin, Utah: Oil and gas well pad emissions compared to ambient air composition, Atmos. Chem. Phys., 14(20), 10977–10988, doi:10.5194/acp-14-10977-2014, 2014.

Watson, J. G., Chow, J. C. and Fujita, E. M.: Review of volatile organic compound source apportionment by chemical mass balance, Atmos. Environ., 35(9), 1567–1584, doi:10.1016/S1352-2310(00)00461-1, 2001.

Wilde, S. E., Dominutti, P. A., Allen, G., Andrews, S. J., Bateson, P., Bauguitte, S. J. B., Burton, R. R., Colfescu, I., France, J., Hopkins, J. R., Huang, L., Jones, A. E., Lachlan-Cope, T., Lee, J. D., Lewis, A. C., Mobbs, S. D., Weiss, A., Young, S. and Purvis, R. M.: Speciation of VOC emissions related to offshore North Sea oil and gas production, Atmos. Chem. Phys., 21(5), 3741–3762, doi:10.5194/acp-21-3741-2021, 2021.

Winkler, J., Blank, P., Glaser, K., Gomes, J. A. G., Habram, M., Jambert, C., Jaeschke, W., Konrad, S., Kurtenbach, R., Lenschow, P., Lörzer, J. C., Perros, P. E., Pesch, M., Prümke, H. J., Rappenglueck, B., Schmitz, T., Slemr, F., Volz-Thomas, A. and Wickert, B.: Ground-Based and Airborne Measurements of Nonmethane Hydrocarbons in BERLIOZ: Analysis and Selected Results, in Tropospheric Chemistry, pp. 465–492, Springer Netherlands., 2002.

World Health Organization.: WHO air quality guidelines for Europe, 2nd edition., Air Qual. Guidel., 2000.

Xie, X., Shao, M., Liu, Y., Lu, S., Chang, C. C. and Chen, Z. M.: Estimate of initial isoprene contribution to ozone formation potential in Beijing, China, Atmos. Environ., 42(24), 6000–6010, doi:10.1016/j.atmosenv.2008.03.035, 2008.

Zhao, W., Hopke, P. K. and Karl, T.: Source Identification of Volatile Organic Compounds in Houston, Texas, Environ. Sci. Technol., 38(5), 1338–1347, doi:10.1021/es034999c, 2004.

Zheng, H., Kong, S., Xing, X., Mao, Y., Hu, T., DIng, Y., Li, G., Liu, D., Li, S. and Qi, S.: Monitoring of volatile organic compounds (VOCs) from an oil and gas station in northwest China for 1 year, Atmos. Chem. Phys., 18(7), 4567–4595, doi:10.5194/acp-18-4567-2018, 2018.

Zou, Y., Deng, X. J., Zhu, D., Gong, D. C., Wang, H., Li, F., Tan, H. B., Deng, T., Mai, B. R., Liu, X. T. and Wang, B. G.: Characteristics of 1 year of observational data of VOCs, NOx and O3 at a suburban site in Guangzhou, China, Atmos. Chem. Phys., 15(12), 6625–6636, doi:10.5194/acp-15-6625-2015, 2015.