# Peer review of "Measurement report: Summertime and wintertime VOCs in Houston: Source apportionment and spatial distribution of source origins"

_Atmospheric Chemistry and Physics, 2021_

## Author Comment (AC1)

**Responses to the reviewers**

We express our gratitude to both reviewers for their comments, as well as their critical remarks, which have helped us make this paper more organized and scientifically robust. We authors let the reviewer know that we got some feedback from the handling editor who asked us to submit this paper as the category of "measurement report" instead of "research articles". Following the editor's suggestion, we submitted this manuscript as the category of "measurement report" (please refer to the title, "Measurement report: Summertime and wintertime VOCs in Houston: Source apportion and spatial distribution of source origins"), and thus, this manuscript includes measurement results, but where the implications of atmospheric chemistry and physics are less developed in this manuscript (refer to ACP manuscript types).

**Anonymous Referee #1:**

**General comments:**

**The research about the local VOC properties is basically interesting and fits into the scope of ACP. To be published, however, a dense revision looks required.**

1. **To derive some essential key points of this study looks necessary. This is a study at a certain region (Houston). In this kind of study, readers would like to see why we need to understand the VOC pattern in Houston, meaning that what is similar to/or different from VOC properties in other regions. That is a very important information for making a study about the local air quality suggesting some generalized lessons to the readers.**

We thank the reviewer for this constructive suggestion. The comment above led to a series of revisions throughout different sections of this manuscript as follows:

A series of emission control policies were conducted to improve the air quality of Houston during the last two decades. However, the city's high level of ozone still is an issue that caused the city to be one of the nonattainment areas by the U.S. EPA. The city is among the largest metropolitan areas containing petrochemical and industrial activities around the world (Sadeghi et al., 2020). Hence, studying the air quality management of Houston could be an experimental lab for policymakers who want to alleviate ozone pollution in urban regions. The problem of ozone air

pollution and the features of the city of Houston encouraged the authors, as the people who are residents of Houston, to conduct this research and take a deeper look to investigate the characteristics of the volatile organic compounds and their emissions sources.

In regard to the reviewer's comments, we made a series of changes to different sections of the manuscript to help the readers having a more clear view of the necessity and findings of this paper. Here is a list of changes we made to this manuscript:

- We clarified how the city's exceeding ozone levels made it a remarkable area for studying volatile organic compounds:

*Houston metropolitan area is well-known for its air quality challenges, and it has been exceeding the U.S. National Ambient Air Quality Standard (NAAQS) for ozone over the last two decades. Some particular features are affecting Houston's ozone problems.* The Houston metropolitan area has some of the largest anthropogenic emission sources of atmospheric pollutants in the United States (Song et al., 2021). The Houston Ship Channel area, for example, is affected by high rates of numerous pollutants from petrochemical and industrial facilities (Port Houston, 2019). *Therefore, the city is a remarkable area for characterizing the emissions of organic compounds and their impacts on ozone formation in urban areas.*

- We intended to investigate the patterns of volatile organic compounds and their emissions one decade after Buzcu and Fraser (2006) and Leuchner and Rappenglueck (2010) studied the volatile organic compounds at the Ship Channel area. We presented a summary of what they found of the VOCs over Houston:

*Multiple studies have identified these VOC characteristics, their emission sources, and contributions in photochemical processes in several urban areas (Bi et al., 2021; Buzcu and Fraser, 2006; Czader et al., 2008; Czader and Rappenglueck, 2015; Diao et al., 2016; Jobson et al., 2004; Leuchner and Rappenglueck, 2010; Pan et al., 2015). These studies are essential to review the effectiveness of control strategies since any strategy for emission mitigations of VOCs necessitates a quantitative assessment of their emissions and understanding of the source-receptor relationships (Demerjian, 2000).* They showed that the main sources of anthropogenic emissions are industrial processes, including those of crude oil and liquefied petroleum gas (LPG),

*hereinafter defined as oil and natural gas (ONG), gasoline transport and storage, vehicle combustion, and the manufacturing production of commercial goods (Piccot et al., 1992).*

- We then explained what was not studied in previous works that our study has taken into account about the VOCs of Houston:

***Previous studies showed that the emissions of VOCs in Houston are mainly from the industrial sectors of the Ship Channel, where they depend on meteorology and transport to urban and rural areas. However, the transient nature of the campaign differences in measurement time periods limited these studies ability to make quantitative comparisons between different seasons (Na and Kim, 2001), and their investigation overlooked the variability of the VOCs concentrations and their photochemical reactivity over Houston (An et al., 2014; Guo et al., 2014; Pan et al., 2015; Baudic et al., 2016). In addition, these studies lacked the consideration of the different formation potentials of speciated VOC, which is necessary to address the causes of ozone episodes and mitigate urban air pollution.***

- At the end of the introduction, we added a new paragraph explaining what this study wants to present. This paragraph clarified what we conducted in this study and helps the readers to know about what questions the authors plan to answer.

***This study aims to (1) explore the characterization of VOCs measurements and their emission sources in two distinct seasons of summer and winter in the Houston Ship Channel. ; (2) elucidate the removal processes of the VOCs and propensity of the individual VOCs to ozone formation potential (OFP). Leuchner and Rappenglueck (2010) provided a categorical view about the wind direction frequency for emission sources, but this paper (3) goes beyond by studying the geographical origins of the air masses coming from the larger Houston metropolitan area.***

- Last two new paragraphs that we have added to the introduction clarify the motivations of this research.

2. **Please add some more efforts to have a better organization. It is difficult to find a main flow of this work (A lot of findings are scatterred so it is hard to catch what is important.**

- Thanks. This is the list of modifications to prepare the organization of the manuscript to help the readers understand the flow of this work:

  - Introduction: The contents were modified as we mentioned in the previous part.
  - Materials and Methods:

    - Section 2.1, Figure 2 and its discussion were moved to the results and discussion, section 3.1.

    - Section 2.2, more explanations were added to the methods of source apportionment. A list of references that recently deployed the PMF model to study the VOCs was added. We also clarified how we performed the PCA and PMF in this study:

      *For the PMF solution to have physically meaningful results, the number of factors need to represent realistic sources that contribute to the species in matrix $x_{ij}$. We, therefore, performed a multivariate statistical technique of principal component analysis (PCA) to choose a robust number of factors (Sadeghi et al., 2020).*

  - Results and Discussion

    - Section 3.1, the title was changed to "Data overview and general seasonal characteristics of VOCs measured at Ship Channel". We reviewed this section and revised it following the reviewer's request. We compared the results of this study with the results of TRAMP field campaign in summer 2006 (Leuchner and Rappenglueck, 2010). A new figure comparing the results of that campaign with our study was added to the supplement document. Figure 2 and discussion of the graphs were moved into this section.

    - Section 3.2, a deeper discussion with previous findings were included. The results of source apportionment were linked to previous works conducted over Houston (Buzcu and Fraser, 2006; Leuchner and Rappenglueck, 2010).

3. **Please include deeper discussion with previous findings. It is hard to see what is the specialty of study and how this work has the discriminated merit compared to previous VOC works.**

We explored the characteristics of the VOCs in two distinct seasons and discussed the seasonal variations of different groups of VOCs in AutoGc stations around the Houston Ship Channel area. The concentrations of alkanes, alkenes, and aromatics were compared with the measurements of TRAMP field campaign in summer 2006 (Leuchner and Rappenglueck, 2010) in sec. 3.1. Compared to the results of TRAMP field campaign, we explained how the alkane compounds changed since 2006:

*During the TRAMP field campaign in summer 2006, the mixing ratios of alkanes had the highest contributions to the total VOCs (Leuchner and Rappenglueck, 2010). The total mixing ratios of the group of alkanes considered in Table 1 was 79.82 ppbC. Similar to the measurements of summer 2006, alkanes had the highest contributions to the total VOCs so that they were the dominant group of VOCs throughout the last decade. Although the alkane levels utilized for source characterizations during TRAMP field campaign were measured at the Moody Tower on the University of Houston campus, approximately 10 km west from the Houston Ship Channel, the differences of alkanes mixing ratios between TRAMP field campaign and summertime measurements of our study indicate emissions of oil and natural gas VOCs have declined substantially since 2006 (Fig. S1).*

We also explored how the concentrations of alkenes and aromatics changed during the last 15 years:

*Aromatics were another class of VOCs in which a drastic reduction of mixing ratios was observed from 20.04 ppbC to 1.90 ppbC in summer. Unlike groups of alkanes and aromatics which vastly changed since 2006, the average mixing ratios of alkene compounds showed a mild variation from 12.05 ppbC to 9.01 ppbC. As shown in Fig. S1, ethylene and propylene are two major alkenes with the highest potential contributions to the formation of ozone that revealed a relatively constant level of concentration.*

The results of our source apportionment were presented in sec. 3.2. The seasonal characteristics of the VOCs emissions were investigated, and the main anthropogenic emissions were compared with previous studies (Buzcu and Fraser, 2006; Leuchner and Rappenglueck, 2010).

[Figure]

Fig. S1. Average concentrations of major compounds for Alkanes (top), Alkenes (middle), and Aromatics (bottom) during summer and winter 2018 and summer 2006 (Leuchner and Rappenglueck, 2010).

**4. The research motivation was not clarified in the introduction.**

Following the reviewer's first comment, a new paragraph was added to the last part of the introduction, and the motivation of the study was presented to the readers. Here are the main elements that motivated us to conduct this study, and we presented in the introduction: The time period of previous campaign measurements limited our understandings of the variations of characteristics of the VOCs in different seasons. As we have clarified in the introduction:

*This study aims to (1) explore the characterization of VOCs measurements and their emission sources in two distinct seasons of summer and winter in the Houston Ship Channel. ; (2) elucidate the removal processes of the VOCs and propensity of the individual VOCs to ozone formation potential (OFP). Leuchner and Rappenglueck (2010) provided a categorical view about the wind direction frequency for emission sources, but this paper (3) goes beyond by studying the geographical origins of the air masses coming from the larger Houston metropolitan area.*

**5. Several statements were provided without references. Please proofread whole manuscript again and reinforce the references.**

The manuscript was proofread and different references were added in the sections of the manuscript.

**6. Results are too peripheral. I do not know how to apply/generalize these results for the advanced understanding of VOC chemistry.**

This study is considered to be an example of a measurement report study at a metropolitan area with significant levels of emissions. We defined three objectives for this study: seasonal characteristics of VOCs measurements and their emission sources, photochemical reactivity of the organic compounds, and their contributions to ozone formation and spatial distributions of the emissions factors around the industrial region. Also, a comparison of the results of this study with a previous field measurement campaign in summer 2006 would help the readers to have a clear outlook about the chemistry of VOCs over Houston.

**7. I agree that the results are interesting and informative for the local air quality study. But the dataset is not very something new and most analysis is so qualitative based on simple PMF and backtrajectory calculation (As author mentined in the manuscript, results usually support the**

**previous findings. New lessons or arguments cannot be found much). The final decision is not sure but therefore it seems better to submit this manuscript to other journals if serious revision is not guaranteed (For example, Atmopsheric Environment)**

Actually, this handling editor asked us to submit this paper as the category of the ACP, "measurement report" instead of research article". Following the editor's suggestion, the type of manuscript this paper was considered as the category as "measurement report" in Atmospheric Chemistry and Physics (ACP) (please refer to the title, "Measurement report: Summertime and wintertime VOCs in Houston: Source apportionment and spatial distribution of source origins"). Analysis of the measurements may include model results and conclusions of more limited scope than in research articles.

**Specific comments:**

**Line 78-80: Not well connected to the previous statements (Suddenly VOC inflence is connected to the VOC source)**

Thanks. Revised. This sentence was edited and moved to the next paragraph.

*These studies are essential to review the effectiveness of control strategies since any strategy for emission mitigations of VOCs necessitates a quantitative assessment of their emissions and understanding of the source-receptor relationships (Demerjian, 2000).*

**line 94-103: It seems that there are already deep studies about VOCs at Houston area. Considering past findings, what do we need to know more?**

**What is the missing points in the previous works. There things are not shown in this introduction parts.**

We appreciate the reviewer's comment. As the reviewer suggested, the authors revised the sections of introduction and results and discussion according to this comment. In general, despite all the regulations and pollution control policies that have decreased the city's emissions of organic compounds during the last two decades, Houston is still dealing with the impacts of surface ozone, and this area has been classified as an ozone nonattainment area by U.S. EPA.

Following the reviewer's general comment #1, we edited the introduction of this paper to make the general lessons of the manuscript more transparent for the readers. We explained what previous studies have found and what we are seeking to attain. Overall, we think our study focused more on the temporal and spatial variability of the VOC compounds over Houston. We investigated the temporal variations of characteristics of VOCs in two different seasons of summer and winter and discussed how the profiles of emissions sources of VOCs change by these seasons. We also explored the spatial distributions of the organic compounds by studying the geographical origins of VOCs emissions affecting the Ship Channel area.

Following one of the reviewer's general comments to explain the motivations of this study, we clarified what motivated the authors to study the characteristics of the organic compounds over Ship Channel.

**Line 147-151: A reference is required about the shorter lifetime of atmospheric alkene/aromatics than alkan + longer lifetime of ethane and propane.**

Revised. Section 3.1 of the results and discussion was modified following the reviewer's general comments. So, this sentence was modified and a reference is included as follows:

*The more rapid reaction of the alkenes and aromatics with radicals would likely cause their seasonal differences to be less variant (Atkinson and Arey, 2003).*

**Line 149-150: => Differences of alkane concentrations**

Corrected. Thanks.

**line 247-248: This should be confirmen with some data analysis in Houston. Different from the convective mixing and photochemical activity (discussed after this statement), energy demand pattern can has a larger regional difference, so it is hard to generalized simply without the specific clue.**

Thanks. We have revised these sentences as follows:

*There are several factors such as emission sources, convective mixing, and photochemical reactions which caused the variations of concentrations between two seasons. Differences in alkane concentrations between the summertime and wintertime could be explained by a mixture of these factors (Atkinson and Arey, 2003). Total Texas hourly electricity load indicates there is*

*a seasonal peak reflecting fluctuations in energy demand in the afternoon occurs when households and businesses increase the use of electricity for air conditioning on hot days (EIA, 2019). Although higher demands of energy usage during the summer months could result in higher levels of VOCs emissions, higher planetary boundary layer and stronger vigorous turbulent mixing, which intense the daytime convective boundary layer likely led to reduced summertime concentrations compared to wintertime (Velasco et al., 2008).*

**Line 300-304: Important information to compare between C2-C5 and C8-C10 isomers. But how to obtain these OH reaction rate constant is not clear here. Any related figure or table is not found in this manuscript.**

Previous studies investigated the main removal pathway for VOCs (alkane, alkene, and aromatics), which occurs through chemical oxidation by OH radicals (Atkinson, 2000; Koppmann, 2008). We have used the reaction rate coefficients from the study of Atkinson and Arey (2003) and listed them in Table 1. Following the reviewer's comment, we revised these lines, addressed Table 1, and added the reference to show where we found these reaction rates values.

**line 307: 'in Figs. 4 and 5.' => 'in Figs. 4 and 5, for summertime and wintertime, respectively'**

Thanks. Revised.

**line 308: What are 3 ONG species factors? (Factor 1, 3, and which? Clearly indicate the factors)**

Changed (… three ONG species factors, ONG long-lived, fuel evaporation, and ONG short-lived).

**line 309-310: How to justify this statement?**

Revised. The whole paragraph was revised. More statements were added to explain the diurnal variations of the resolved factors in summertime and winter. We clarified the diurnal variations of the resolved factors in associate with the degradation rate of their dominant compounds:

*Because of varying rates of oxidation of the vapor-phase organic compounds, some highly reactive species are depleted at a more rapid rate than less-reactive compounds. With the transport of emissions from distant sources, atmospheric oxidation removes the reactive compounds and the remaining fraction contains mostly the less-reactive species, such as ethane*

*and propane (De Gouw et al., 2005). Therefore, having longer lifetimes, ethane and propane tend to stay and accumulate in the atmosphere and may cause an overestimation of oil and natural gas emissions which is determined based on the concentrations of less-reactive alkanes (Buzcu and Fraser, 2006).*

**line 307-314: Most interpretations are not connected to clues. At least the references should be added.**

Thanks. Revised. The paragraph was revised and references were added.

**line 319-356: Please write this part again matching to the findings in Fig. 4 and 5. Different from Factor 1, 2, and, 3, analysis about factor 4, 5, 6 looks only based on Fig. 3. There is no examination about the diurnal cycle. Organization of paragraphs look problematic. This part should be significantly improved.**

Thanks. Revised. The entire paragraph was revised.

**line 344-346: What is the clue to support this statement? I cannot figure out this, even based on Fig. 3.**

The first five factors in summertime and wintertime show similar characteristics in terms of high percentages of specific compounds, i.e., higher contributions of ethane and propane in factor 1 during both summer and winter. Factor 6 and 7 in the summertime are independent emission sources that have not been resolved in wintertime. Similarly, factor 6 was independently resolved for wintertime. The statement was modified as below:

*Figures 4 and 5 show the factors of one to five posed similar characteristics in both the summertime and the wintertime, but factors 6 and 7 for the summertime and factor 6 for the wintertime presented different features.*

**line 393-395: How to justify this interpretation?**

Thanks. We corrected our previous claim and modified the sentence as follows:

*The ratio of benzene to toluene notably increased from the wintertime (0.95 ppbC) to the summertime (3.10 ppbC). The ratio of ethylbenzene/m.p-xylene, two compounds with less discrepancy in their lifetime changed only slightly between the two periods (0.19 for summer*

*and 0.21 for winter), suggesting the chemical oxidation of reactive compounds accelerate the degradation removal of organic compounds in Houston Ship Channel during the summer.*

**line 486-493: Overall summary is so ambiguous. Specific lessons to make summer-winter difference are necessary.**

Response: Modified. We edited the summary and focused on the main findings of this measurement report study.

**Whole results: The reason only focusing on summer and winter is not clear, while the full year measurement data exist. Is there any specific reason not to see the spring and autumn. It is also worth to investigate other seasons (For example, Hurricane effect is also huge in autumn).**

Most of Houston is located on Western Gulf coastal grassland, subtropical areas of the south of Texas. Hence the climate of Houston is a subtropical climate where the temperature usually stays high throughout the year and cools down for only a few months. So there are really only 2 seasons here: summer and winter. Houston experiences longer summer since the northeast of Texas is near the equator. For example, in the year 2018, three months of winter showed an average of 55º F while the other nine months showed an average of 75º F. and the average of five months was above 80 º F (May to Sep). The other reason we focused on summer and winter was to see the contrast between summertime with a pronounced level of ozone "ozone season" and winter where the ozone problem is less critical.

**National Weather Service**

**https://www.weather.gov/hgx/climate_graphs_iah#2018**

Thank you again for your detailed remarks; we have adapted the manuscript as suggested.

**References**

An, J., Zhu, B., Wang, H., Li, Y., Lin, X. and Yang, H.: Characteristics and source apportionment of VOCs measured in an industrial area of Nanjing, Yangtze River Delta, China, Atmos. Environ., 97, 206–214, 2014.

Atkinson, R.: Atmospheric chemistry of VOCs and NO(x), Atmos. Environ., 34(12–14), 2063–2101, doi:10.1016/S1352-2310(99)00460-4, 2000.

Atkinson, R. and Arey, J.: Atmospheric Degradation of Volatile Organic Compounds, Chem. Rev., doi:10.1021/cr0206420, 2003.

Baudic, A., Gros, V., Sauvage, S., Locoge, N., Sanchez, O., Sarda-Estève, R., Kalogridis, C., Petit, J.-E., Bonnaire, N., Baisnée, D. and others: Seasonal variability and source apportionment of volatile organic compounds (VOCs) in the Paris megacity (France), Atmos. Chem. Phys., 16(18), 11961–11989, 2016.

Bi, S., Kiaghadi, A., Schulze, B. C., Bernier, C., Bedient, P. B., Padgett, J. E., Rifai, H. and Griffin, R. J.: Simulation of potential formation of atmospheric pollution from aboveground storage tank leakage after severe storms, Atmos. Environ., 248, 118225, doi:10.1016/j.atmosenv.2021.118225, 2021.

Buzcu, B. and Fraser, M. P.: Source identification and apportionment of volatile organic compounds in Houston, TX, Atmos. Environ., 40(13), 2385–2400, doi:10.1016/j.atmosenv.2005.12.020, 2006.

Czader, B. H. and Rappenglueck, B.: Modeling of 1,3-butadiene in urban and industrial areas, Atmos. Environ., 102, 30–42, doi:10.1016/j.atmosenv.2014.11.039, 2015.

Czader, B. H., Byun, D. W., Kim, S. T. and Carter, W. P. L.: A study of VOC reactivity in the Houston-Galveston air mixture utilizing an extended version of SAPRC-99 chemical mechanism, Atmos. Environ., doi:10.1016/j.atmosenv.2008.01.039, 2008.

Demerjian, K. L.: A review of national monitoring networks in North America, Atmos. Environ.,

34(12–14), 1861–1884, 2000.

Diao, L., Choi, Y., Czader, B., Li, X., Pan, S., Roy, A., Souri, A. H., Estes, M. and Jeon, W.: Discrepancies between modeled and observed nocturnal isoprene in an urban environment and the possible causes: A case study in Houston, Atmos. Res., 181, 257–264, doi:10.1016/j.atmosres.2016.07.009, 2016.

De Gouw, J. A., Middlebrook, A. M., Warneke, C., Goldan, P. D., Kuster, W. C., Roberts, J. M., Fehsenfeld, F. C., Worsnop, D. R., Canagaratna, M. R., Pszenny, A. A. P. and others: Budget of organic carbon in a polluted atmosphere: Results from the New England Air Quality Study in 2002, J. Geophys. Res. Atmos., 110(D16), 2005.

Guo, J., Tilgner, A., Yeung, C., Wang, Z., Louie, P. K. K., Luk, C. W. Y., Xu, Z., Yuan, C., Gao, Y., Poon, S. and others: Atmospheric peroxides in a polluted subtropical environment: seasonal variation, sources and sinks, and importance of heterogeneous processes, Environ. Sci. Technol., 48(3), 1443–1450, 2014.

Jobson, B. T., Berkowitz, C. M., Kuster, W. C., Goldan, P. D., Williams, E. J., Fesenfeld, F. C., Apel, E. C., Karl, T., Lonneman, W. A. and Riemer, D.: Hydrocarbon source signatures in Houston, Texas: Influence of the petrochemical industry, J. Geophys. Res. D Atmos., 109(24), 1–26, doi:10.1029/2004JD004887, 2004.

Koppmann, R.: Volatile Organic Compounds in the Atmosphere, John Wiley Sons [online] Available from:
https://books.google.com/books?hl=en&lr=&id=uUN3lFs_pgoC&oi=fnd&pg=PP2&dq=volatile +organic+chemistry+in+the+atmosphere&ots=HGf3q0HFCQ&sig=R7sUo5q9W3BV7fyJXP8K 7j_XvqI#v=onepage&q=volatile organic chemistry in the atmosphere&f=false (Accessed 17 June 2021), 2008.

Leuchner, M. and Rappenglueck, B.: VOC source-receptor relationships in Houston during TexAQS-II, , doi:10.1016/j.atmosenv.2009.02.029, 2010.

Na, K. and Kim, Y. P.: Seasonal characteristics of ambient volatile organic compounds in Seoul, Korea, Atmos. Environ., 35(15), 2603–2614, 2001.

Pan, S., Choi, Y., Roy, A., Li, X., Jeon, W. and Souri, A. H.: Modeling the uncertainty of several

VOC and its impact on simulated VOC and ozone in Houston, Texas, Atmos. Environ., 120, 404–416, doi:10.1016/j.atmosenv.2015.09.029, 2015.

Piccot, S. D., Watson, J. J. and Jones, J. W.: A global inventory of volatile organic compound emissions from anthropogenic sources, J. Geophys. Res., 97(D9), 9897–9912, doi:10.1029/92JD00682, 1992.

Port Houston: Overview - Port Houston, [online] Available from: https://porthouston.com/about-us/ (Accessed 17 June 2021), 2019.

Sadeghi, B., Choi, Y., Yoon, S., Flynn, J., Kotsakis, A. and Lee, S.: The characterization of fine particulate matter downwind of Houston: Using integrated factor analysis to identify anthropogenic and natural sources, Environ. Pollut., 262, 114345, doi:10.1016/j.envpol.2020.114345, 2020.

Song, S. K., Choi, Y. N., Choi, Y., Flynn, J. and Sadeghi, B.: Characteristics of aerosol chemical components and their impacts on direct radiative forcing at urban and suburban locations in Southeast Texas, Atmos. Environ., 246, 118151, doi:10.1016/j.atmosenv.2020.118151, 2021.

---

## Author Comment (AC2)

**Responses to the reviewers**

We express our gratitude to both reviewers for their comments, as well as their critical remarks, which have helped us make this paper more organized and scientifically robust. We authors let the reviewer know that we got some feedback from the handling editor who asked us to submit this paper as the category of "measurement report" instead of "research articles". Following the editor's suggestion, we submitted this manuscript as the category of "measurement report" (please refer to the title, "Measurement report: Summertime and wintertime VOCs in Houston: Source apportion and spatial distribution of source origins") and thus, this manuscript includes measurement results, but where the implications of atmospheric chemistry and physics are less developed in this manuscript (refer to ACP manuscript types).

**Anonymous Referee #2:**

**General comments:**

**This manuscript addressed source apportionment of VOCs in Houston, a petrochemistry condensed region in US, by using measurement data in 2018. It reads a quite routine source apportionment report, I regret to suggest a rejection of current version due to three points:**

1. **The current version discussed only the data obtain in their monitoring site, with some discussion on roles of VOCs chemsitry and transportation. I consider that an evaluation on trends of VOCs levels, chemical compositions and source, or features of source chenges for petrochemical industries in US ( Houston as an example) would be of more interest to community rather than a local study.**

Yes. That was a critical comment the reviewer mentioned. The authors clarified the motivations of this study to help the readers what encouraged us to conduct this research. We do think earlier studies showed the emissions of the VOCs over the Houston industrial region (Buzcu and Fraser, 2006; Leuchner and Rappenglueck, 2010). However, there are still some factors needed to be taken into account for an enhanced understanding of the VOCs of Houston. The variation patterns of VOCs depend on anthropogenic emissions, photochemical reactivity, and meteorology. Previous

studies showed that the emissions of VOCs in Houston are mainly from the industrial sectors of the Ship Channel, but the short period of campaign measurements limited those studies' ability to explore the seasonal variations of the anthropogenic and biogenic source emissions. We also looked (Na and Kim, 2001) and their investigation overlooked the variability of the VOCs concentrations and their photochemical reactivity over Houston (An et al., 2014; Guo et al., 2014; Pan et al., 2015; Baudic et al., 2016). In addition, these studies lacked the consideration of the different formation potentials of speciated VOC, which is necessary to address the causes of ozone episodes and mitigate urban air pollution.

2. **The methods used in the MS, PCA, PMF, OFP, ratio analysis and backward trajectory are sort of routine. And the dataset is for 2018, the reviewer didnot see measurements on OVOCs which cuold be important for petrochemical emissions. Therefore from methodological perspective, I didnot found see something new for VOCs source understanding.**

This study has tried to explore the seasonal characteristics of VOCs measurements, their emissions and contributions to ozone formation, and geographical locations in the industrial area of a metropolitan after the last campaign of VOCs characteristics in summer 2006. We used the available data of VOCs measurements in summer and winter and followed the editor's suggestion as to classify this manuscript as a "measurement report" instead "research articles". However, as we can do, we significantly modified and added many subsections to satisfy the need to explore the scientific points (please also refer to our responses to reviewer 1 and the revised manuscript with colored highlights). We hope the revised draft of this manuscript clearly clarifies the findings of our study.

3. **There is an important issue to discuss with the authors. The authors showed quite string chemical oxidation processes by using ratios of VOCs, e.g. Benzene/Toluene, and etc, this clearly means that conventional PCA and PMF could not be deployed for source apportionment in Houston, the authors needs to use chemical-loss correction to do reliable source apportionment. This problem is not discussed in the current version.**

The authors would not argue on capabilities of the PMF model since it is a well-regarded source apportionment technique that provides supportive and reliable information that enhances our understanding of the atmospheric chemistry of organic compounds in numerous recent studies

(e.g., Baudic et al., 2016; Buzcu-Guven and Fraser, 2008; Buzcu and Fraser, 2006; Koss et al., 2020; Leuchner and Rappenglueck, 2010; Li et al., 2019; Liu et al., 2020; Pernov et al., 2021; Pollack et al., 2021; Sinha and Sinha, 2019; Song et al., 2019; Verreyken et al., 2021).

However, we agree with the point the reviewer mentioned and are aware of the fact that the PMF algorithm treats the measured organic compounds as inert species, which addresses some limitations to the interpretation of the results. Hence, as the reviewer suggested, we discussed in the manuscript that processes such as mass production and atmospheric removal could cause a single emission source to appear as multiple factors or causing oxidized species from multiple emission sources to be grouped together and increase the uncertainty of the resolved source profiles (Sauvage et al., 2009; Wang et al., 2013; Yuan et al., 2012).

Thank you again for your detailed remarks; we have adapted the manuscript as suggested.

**References**

Baudic, A., Gros, V., Sauvage, S., Locoge, N., Sanchez, O., Sarda-Estève, R., Kalogridis, C., Petit, J.-E., Bonnaire, N., Baisnée, D. and others: Seasonal variability and source apportionment of volatile organic compounds (VOCs) in the Paris megacity (France), Atmos. Chem. Phys., 16(18), 11961–11989, 2016.

Buzcu-Guven, B. and Fraser, M. P.: Comparison of VOC emissions inventory data with source apportionment results for Houston, TX, Atmos. Environ., 42(20), 5032–5043, 2008.

Buzcu, B. and Fraser, M. P.: Source identification and apportionment of volatile organic compounds in Houston, TX, Atmos. Environ., 40(13), 2385–2400, doi:10.1016/j.atmosenv.2005.12.020, 2006.

Koss, A. R., Canagaratna, M. R., Zaytsev, A., Krechmer, J. E., Breitenlechner, M., Nihill, K. J., Lim, C. Y., Rowe, J. C., Roscioli, J. R., Keutsch, F. N. and others: Dimensionality-reduction techniques for complex mass spectrometric datasets: application to laboratory atmospheric organic oxidation experiments, Atmos. Chem. Phys., 20(2), 1021–1041, 2020.

Leuchner, M. and Rappenglueck, B.: VOC source-receptor relationships in Houston during TexAQS-II, , doi:10.1016/j.atmosenv.2009.02.029, 2010.

Li, B., Ho, S. S. H., Gong, S., Ni, J., Li, H., Han, L., Yang, Y., Qi, Y. and Zhao, D.: Characterization of VOCs and their related atmospheric processes in a central Chinese city during severe ozone pollution periods, Atmos. Chem. Phys., 19(1), 617–638, doi:10.5194/ACP-19-617-2019, 2019.

Liu, Y., Song, M., Liu, X., Zhang, Y., Hui, L., Kong, L., Zhang, Y., Zhang, C., Qu, Y., An, J., Ma, D., Tan, Q. and Feng, M.: Characterization and sources of volatile organic compounds (VOCs) and their related changes during ozone pollution days in 2016 in Beijing, China, Environ. Pollut., 257, 113599, doi:10.1016/J.ENVPOL.2019.113599, 2020.

Pernov, J. B., Bossi, R., Lebourgeois, T., Nøjgaard, J. K., Holzinger, R., Hjorth, J. L. and Skov,

H.: Atmospheric VOC measurements at a High Arctic site: Characteristics and source apportionment, Atmos. Chem. Phys., 21(4), 2895–2916, doi:10.5194/ACP-21-2895-2021, 2021.

Pollack, I. B., Helmig, D., O'Dell, K. and Fischer, E. V: Seasonality and Source Apportionment of Nonmethane Volatile Organic Compounds at Boulder Reservoir, Colorado, Between 2017 and 2019, J. Geophys. Res. Atmos., 126(9), doi:10.1029/2020JD034234, 2021.

Sauvage, S., Plaisance, H., Locoge, N., Wroblewski, A., Coddeville, P. and Galloo, J. C.: Long term measurement and source apportionment of non-methane hydrocarbons in three French rural areas, Atmos. Environ., 43(15), 2430–2441, 2009.

Sinha, B. and Sinha, V.: Source apportionment of volatile organic compounds in the northwest Indo-Gangetic Plain using a positive matrix factorization model, Atmos. Chem. Phys, 19, 15467–15482, doi:10.5194/acp-19-15467-2019, 2019.

Song, M., Liu, X., Zhang, Y., Shao, M., Lu, K., Tan, Q., Feng, M. and Qu, Y.: Sources and abatement mechanisms of VOCs in southern China, Atmos. Environ., 201, 28–40, 2019.

Verreyken, B., Amelynck, C., Schoon, N., Müller, J.-F., Brioude, J., Kumps, N., Hermans, C., Metzger, J.-M., Colomb, A. and Stavrakou, T.: Measurement report: Source apportionment of volatile organic compounds at the remote high-altitude Ma{\"\i}do observatory, Atmos. Chem. Phys., 21(17), 12965–12988, 2021.

Wang, H. L., Chen, C. H., Wang, Q., Huang, C., Su, L. Y., Huang, H. Y., Lou, S. R., Zhou, M., Li, L., Qiao, L. P. and others: Chemical loss of volatile organic compounds and its impact on the source analysis through a two-year continuous measurement, Atmos. Environ., 80, 488–498, 2013.

Yuan, B., Shao, M., De Gouw, J., Parrish, D. D., Lu, S., Wang, M., Zeng, L., Zhang, Q., Song, Y., Zhang, J. and others: Volatile organic compounds (VOCs) in urban air: How chemistry affects the interpretation of positive matrix factorization (PMF) analysis, J. Geophys. Res. Atmos., 117(D24), 2012.